# Attack-Resilient Image Watermarking Using Stable Diffusion

**Lijun Zhang**
University of Massachusetts
Amherst, MA 01003-9264
lijunzhang@cs.umass.edu

**Xiao Liu**
University of Massachusetts
Amherst, MA 01003-9264
xiaoliu1990@umass.edu

**Antoni Viros Martin**
IBM Watson
Yorktown Heights, NY 10598
aviros@ibm.com

**Cindy Xiong Bearfield**
Georgia Institute of Technology
Atlanta, GA 30332
cxiong@gatech.edu

**Yuriy Brun**
University of Massachusetts
Amherst, MA 01003-9264
brun@cs.umass.edu

**Hui Guan**
University of Massachusetts
Amherst, MA 01003-9264
huiguan@cs.umass.edu

## Abstract

Watermarking images is critical for tracking image provenance and proving ownership. With the advent of generative models, such as stable diffusion, that can create fake but realistic images, watermarking has become particularly important to make human-created images reliably identifiable. Unfortunately, the very same stable diffusion technology can remove watermarks injected using existing methods. To address this problem, we present ZoDiac, which uses a pre-trained stable diffusion model to inject a watermark into the trainable latent space, resulting in watermarks that can be reliably detected in the latent vector even when attacked. We evaluate ZoDiac on three benchmarks, MS-COCO, DiffusionDB, and WikiArt, and find that ZoDiac is robust against state-of-the-art watermark attacks, with a watermark detection rate above $98\%$ and a false positive rate below $6.4\%$, outperforming state-of-the-art watermarking methods. We hypothesize that the reciprocating denoising process in diffusion models may inherently enhance the robustness of the watermark when faced with strong attacks and validate the hypothesis. Our research demonstrates that stable diffusion is a promising approach to robust watermarking, able to withstand even stable-diffusion–based attack methods. ZoDiac is open-sourced and available at https://github.com/zhanglijun95/ZoDiac.

## 1 Introduction

Digital image watermarking, a technique for subtly embedding information within digital images, has become increasingly crucial and beneficial in the context of content protection and authenticity verification [36, 10, 5, 4]. The advance in generative AI technologies [28, 30, 29], such as stable diffusion, further underscores the need for watermarking solutions to distinguish between AI-generated and human-created images.

Given an existing image, many methods have been proposed to embed a watermark that is invisible and robust to watermark removal attacks. Conventional watermarking strategies have employed various methods, such as embedding information in texture-rich regions [3], manipulating the least significant bits [43], or utilizing the frequency domain [21]. The emergence of deep learning has introduced neural network (NN)-based watermarking methods [53, 24, 35, 48, 47, 19], which have shown promise in achieving high invisibility and robustness against traditional attacks, such as adding Gaussian noise or applying JPEG compression.

38th Conference on Neural Information Processing Systems (NeurIPS 2024).

Unfortunately, the recent advent of powerful image generation techniques can circumvent existing watermarking methods. The most recent work [51] shows that stable diffusion can be used in a watermark removal attack, and none of the existing watermarking techniques are robust enough to that attack. Figure 1 shows that the watermark detection rate (WDR) of six existing watermarking methods (DwtDct, DwtDctSvd [9], RivaGAN [48], SSL [16], CIN [25], StegaStamp [35]) before and after the stable-diffusion-based watermark removal attack [51] drops from 79%–100% before attack to only 0%–48% after attack on the MS-COCO dataset [23].

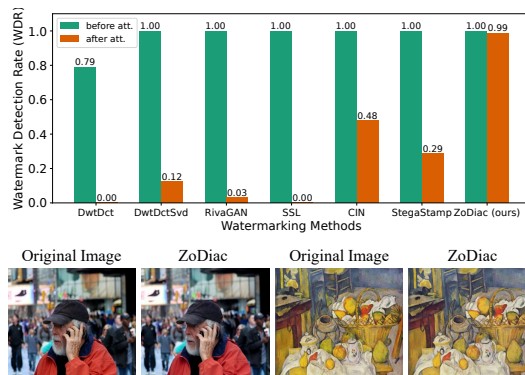

Figure 1: The watermark detection rate of existing methods and our ZoDiac before and after the diffusion-based attack *Zhao23* [51]. Two example images show that ZoDiac's watermarks are perceptually invisible.

To address the problem, we propose a novel stable-diffusion–based watermarking framework called ZoDiac. ZoDiac takes as input an existing image and uses a pre-trained stable diffusion model to inject into the image a watermark that can be reliably detected even when attacked. The rationale behind ZoDiac is that a pre-trained stable diffusion maps a latent vector into an image and that *more than one latent vector can be mapped to perceptually the same image*. Given an existing image, ZoDiac identifies a latent vector that contains a watermark pattern and can also be mapped to the same image using a pre-trained stable diffusion model. ZoDiac further allows mixes of the original image and the watermarked image to enhance image quality with a minor influence on watermark robustness. To detect watermarks, ZoDiac applies a diffusion inversion process that remaps the image to the latent vector and then detects the watermark pattern in the latent vector by statistical test. Recent trend [11, 15, 42] has used diffusion models for watermarking, but it can only encode a watermark into synthetically-generated images when they are being generated. By contrast, ZoDiac can watermark existing and real-world images.

ZoDiac has two distinctive features that make it robust and effective. First, ZoDiac injects watermarks in the latent space that stable diffusion models operate on when sampling random noise to generate synthetic images, making the watermarks both invisible and robust to even the most advanced stable-diffusion–based attack method. We hypothesize that ZoDiac exhibits strong robustness because the image generation process of the diffusion model serves as a powerful attack-defense mechanism by default and empirically validate it in §4.3. Second, while existing watermarking methods typically require the training of a dedicated model on a representative training dataset, ZoDiac builds on top of pre-trained stable diffusion models, forgoing the time-consuming training process.

Our main contributions are:

- *ZoDiac* — a novel framework for embedding invisible watermarks into existing images using pre-trained stable diffusion. To the best of our knowledge, ZoDiac is the first watermarking method that is robust to the most advanced generative AI-based watermark removal attack.
- *Empirically Demonstrated Strong Robustness Against Watermark Attacks* — Our evaluation on images from diverse domains, MS-COCO [23], DiffusionDB [41], and WikiArt [27] datasets, shows that ZoDiac is robust against the state-of-the-art watermark attack mechanism, with watermark detection rate (i.e., true positive rate) above $98\%$ and false positive rate below $6.4\%$, outperforming state-of-the-art watermarking methods. ZoDiac also maintains invisible image quality degradation, underscoring its efficacy in achieving robust watermarking with quality preservation (see examples in Figure 1).
- *Robustness Against Combined Attacks* — Prior watermarking method evaluations [42, 51] focused on robustness to only a single attack at a time. We show that in a more realistic scenario, where the attacker can combine multiple attacks, ZoDiac significantly outperforms all existing methods. For example, when combining all attacks other than image rotation, ZoDiac retains a detection rate above $50\%$ while all existing methods fail with a detection rate of $0\%$. Only one of the existing methods, SSL [16], is effective against rotation, but it is ineffective against other attacks. We propose a method for making ZoDiac robust to rotation (above $99\%$), only slightly increasing its false-positive rate (from $0.4\%$ to $3.4\%$) on MS-COCO dataset with appropriate hyperparameter settings.

## 2 Preliminary and Related Work

We first introduce the image watermarking problem and then the diffusion model background necessary to understand ZoDiac. We place our work in the context of the most related research; Appendix A discusses related work in depth.

**The Robust Image Watermarking Problem.** Image watermarking aims to embed a predefined, detectable watermark into an image while ensuring that the watermarked image is similar to the original. This similarity is usually quantified using metrics such as the Peak Signal-to-Noise Ratio (PSNR) and the Structural Similarity Index (SSIM) [40]. Malicious attacks can attempt to remove the watermark, again, without significantly changing the image. A *robust* image watermarking method should be able to detect the watermark even in attacked images.

Both image watermarking and watermark removal attacks have been studied extensively. The development of deep neural networks (DNN) resulted in advanced learning-based watermarking approaches, such as RivaGAN [48], StegaStamp [35], and SSL [15]. These methods train a dedicated DNN model on large-scale representative image datasets to watermark images. Meanwhile, advances in generative AI techniques have produced new attacks based on Variational AutoEncoders [2, 8] and Stable Diffusion [29]. None of the existing watermarking approaches are sufficiently robust to these advanced diffusion-based attacks.

**Diffusion Models and DDIM Inversion.** Diffusion models [31], which both the most sophisticated attacks and our approach build on, are a class of generative AI models that generate high-resolution images. To explain diffusion models, we first explain the forward diffusion process, in which a data point sampled from a real data distribution $x_0 \sim q(x)$ is gradually converted into a noisy representation $x_T$ through $T$ steps of progressive Gaussian noise addition. This transformation yields $x_T$ as an isotropic Gaussian noise, i.e., $x_T \sim \mathcal{N}(\mathbf{0}, \mathbf{I})$. Specifically, a direct generation of $x_t$ from $x_0$ is:

$$x_t = \sqrt{\bar{\alpha}_t} x_0 + \sqrt{1 - \bar{\alpha}_t} \epsilon, \tag{1}$$

where $\bar{\alpha}_t = \prod_{i=0}^{t}(1 - \beta_i)$, $\beta_i \in (0, 1)$ is the scheduled noise variance that controls the step size, and $\epsilon \sim \mathcal{N}(0, \mathbf{I})$.

Diffusion models reverse this forward process, learning to retrieve the original image $x_0$ from the noise $x_T$ by estimating the noise at each step and iteratively performing denoising. The Denoising Diffusion Implicit Model (DDIM) [33] is a prominent denoising method, known for its efficiency and deterministic output. Formally, for each denoising step $t$, a learned noise predictor $\epsilon_\theta$ estimates the noise $\epsilon_\theta(x_t)$ added to $x_0$, leading to an approximate of $x_0$. Then DDIM reintroduces $\epsilon_\theta(x_t)$ to determine $x_{t-1}$:

$$x_{t-1} = \sqrt{\bar{\alpha}_{t-1}} \left( \frac{x_t - \sqrt{1 - \bar{\alpha}_t} \epsilon_\theta(x_t)}{\sqrt{\bar{\alpha}_t}} \right) + \sqrt{1 - \bar{\alpha}_{t-1}} \epsilon_\theta(x_t). \tag{2}$$

In this way, DDIM could deterministically recover the same image $x_0$ from the specified noise $x_T$. DDIM also enables an inversion mechanism [14] that can reconstruct the noise $x_T$ from an image $x_0$. DDIM inversion adheres to the forward diffusion process in Eq. (1), substituting $\epsilon_t$ with $\epsilon_\theta(x_t)$ at each timestep.

We denote the denoising process, that is, the image generation process as $\mathcal{G}$ and its inversion as $\mathcal{G}'$. A stable diffusion model [29] operates in the latent space. We use $\mathbf{Z}_T$ and $\mathbf{Z}_0$ in the latent space to represent the noise $x_T$ and the image $x_0$ respectively. A pre-trained Variational Autoencoder (VAE) carries out the transformation between the noise $x_T$ and its latent vector $\mathbf{Z}_T$ (and also the image $x_0$ and its representation $\mathbf{Z}_0$). Throughout this paper, we refer to $\mathbf{Z}_T$ as the *latent vector*.

## 3 The ZoDiac Watermarking Framework

This section introduces ZoDiac, our novel watermarking technique that uses pre-trained stable diffusion models to achieve watermark invisibility and strong robustness to attacks. The core idea is to *learn* a latent vector that encodes a pre-defined watermark within its Fourier space, and can be mapped by pre-trained stable diffusion models into an image closely resembling the original.

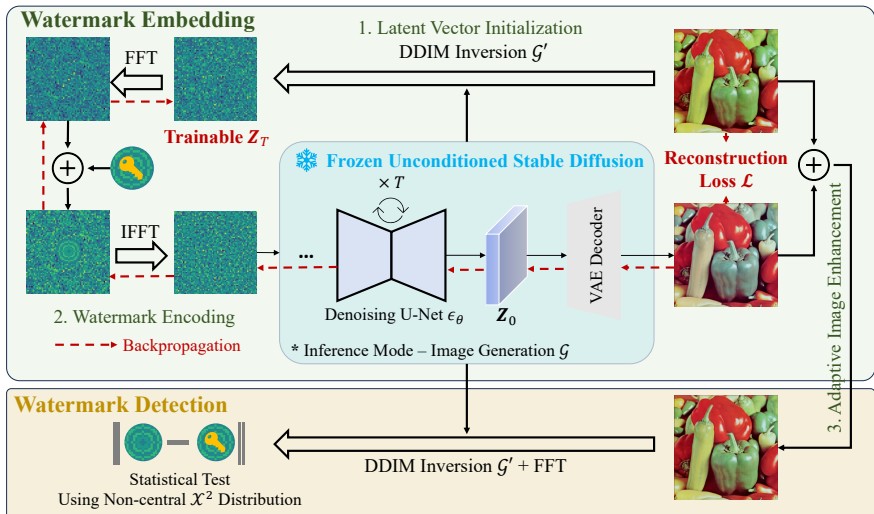

Figure 2: Overview of ZoDiac with watermark embedding and detection phases. There are three major steps in the embedding phase: 1) latent vector initialization, 2) watermark encoding, and 3) adaptive image enhancement. In the detection phase, the watermark is decoded by performing DDIM inversion, Fourier transformation, and statistical testing.

## 3.1 Overview of ZoDiac

Figure 2 illustrates both the watermark embedding and detection phases of the ZoDiac framework. We first explain the high-level idea, and then §3.2 and 3.3 detail each step.

**Watermark Embedding.** Watermark embedding consists of three main steps, *latent vector initialization*, *watermark encoding*, and *adaptive image enhancement*. Algorithm 1 lists the pseudocode for watermark embedding.

The original image $x_0$ first undergoes a DDIM inversion process to identify its latent vector $\mathbf{Z}_T$ (Latent Vector Initialization, Line 1, §3.2.1). ZoDiac then encodes a watermark in the latent vector $\mathbf{Z}_T$ and trains the watermarked latent vector such that a pre-trained stable diffusion model can use it to generate a watermarked image similar to the original image (Watermark Encoding, Lines 2–5, §3.2.2). To encode the watermark into $\mathbf{Z}_T$, ZoDiac converts $\mathbf{Z}_T$ to the Fourier space, encodes a given watermark $\mathbf{W}$, and then transforms it back to the spatial domain prior to being fed into the diffusion model. ZoDiac preserves the visual similarity between the generated image $\hat{x}_0$ and the original image $x_0$ by optimizing the latent vector via a carefully designed reconstruction loss. The diffusion model remains frozen and operates in inference mode during the optimization. The gradient flow during backpropagation is indicated by the dashed

---

**Algorithm 1** ZoDiac-Watermarking

**Require:** original image $x_0$, watermark $\mathbf{W}$
**Require:** pre-trained diffusion model $\mathcal{G}$, and its inversion $\mathcal{G}'$
**Require:** diffusion steps $T$, latent update steps $N$
**Require:** SSIM threshold $s^*$
**Ensure:** watermarked image $\bar{x}_0$
1: $\mathbf{Z}_T \leftarrow \mathcal{G}'(x_0)$
2: **for** $i = 1$ to $N$ **do**
3: $\quad \hat{x}_0 \leftarrow \mathcal{G}(\mathbf{Z}_T \oplus \mathbf{W})$
4: $\quad$ Take gradient descent on $\nabla_{\mathbf{Z}_T}\mathcal{L}(x_0, \hat{x}_0)$ {Eq. (6)}
5: **end for**
6: $\bar{x}_0 = \hat{x}_0 + \gamma(x_0 - \hat{x}_0)$ {Eq. (7)}
7: Search $\gamma^* \in [0, 1]$ s.t. $S(\bar{x}_0, x_0) \geq s^*$ {Eq. (8)}
8: **return** $\bar{x}_0$

---

red arrows in Figure 2. After image generation, ZoDiac mixes the watermarked image $\hat{x}_0$ with the original one $x_0$ to further enhance image quality (Adaptive Image Enhancement, Lines 6–7, §3.2.3).

**Watermark Detection.** Watermark detection detects watermarks. ZoDiac first reconstructs the latent vector of the image under inspection via DDIM inversion, transforms the latent vector into the Fourier space, and then conducts a statistical test to detect a potential watermark (§3.3).

## 3.2 Watermark Embedding

### 3.2.1 Latent Vector Initialization

Finding a good initialization for the latent vector $\mathbf{Z}_T$ allows a stable diffusion model to reproduce the original image $x_0$ rapidly, and is thus critical to reducing the time spent on optimizing that latent vector. ZoDiac employs DDIM inversion to initialize the latent vector $\mathbf{Z}_T = \mathcal{G}'(x_0)$, from which DDIM can then remap to the original image. Empirically, we find that using a randomly initialized latent vector can require hundreds of iterations when remapping, while using an initial latent vector from DDIM inversion converges in dozens of iterations while achieving a higher image quality (see details in Appendix B.1).

### 3.2.2 Watermark Encoding

The watermark encoding step aims to encode a watermark into an image with minimal impact on its visual quality. ZoDiac injects the watermark into the Fourier Space of the initialized latent vector from §3.2.1, and then optimizes the watermarked latent vector to ensure that it can be used to synthesize an image resembling the original. Our fundamental insights are two-fold: (1) injecting watermarks into the latent space of images can effectively improve the watermark robustness thanks to the potential attack-defense capability provided by the diffusion process (see §4.3); (2) injecting watermarks into the Fourier representation of a latent vector helps preserve the quality of the watermarked image (see Appendix B.7).

**Watermark Injection.** ZoDiac injects a concentric ring-like watermark into the Fourier space of the latent vector, leading to a circularly symmetric watermark embedding in the low-frequency domain. Such a pattern is proved to be resistant to geometric transformations and common image distortions [32, 42].

ZoDiac assumes that elements of a watermark $\mathbf{W}$ are generated by randomly sampling from a complex Gaussian distribution, noted as $\mathcal{CN}(0, 1)$. Elements that are of the same distance to the center of the latent vector have the same value, making the watermark "ring-like". Formally, let $\mathcal{F}(\mathbf{Z}_T) \in \mathbb{C}^{ch \times w \times h}$ be the Fourier transformed latent vector, where $ch$ is the number of channels, and $w$ and $h$ are the width and height. Let $p = (i, j)$ be a coordinate, $c = (h/2, w/2)$ be the latent vector's center, and $d(p, c)$ be the Euclidean distance from a coordinate to the center. Each element in the watermark $\mathbf{W} \in \mathbb{C}^{w \times h}$ is:

$$\mathbf{W}_p = w_{\lceil d(p,c) \rceil}, \quad \text{where } w_{\lceil d(p,c) \rceil} \sim \mathcal{CN}(0, 1). \tag{3}$$

ZoDiac also needs a binary mask that indicates the location where the watermark will be applied. Mathematically, let $\mathbf{M} \in \{0, 1\}^{w \times h}$ be a binary mask with a predefined radius $d^*$. Each item in the mask $\mathbf{M}$ is

$$\mathbf{M}_p = \begin{cases} 1, & \text{if } d(p, c) \leq d^*; \\ 0, & \text{otherwise.} \end{cases} \tag{4}$$

Finally, the watermark $\mathbf{W}$ is applied to the Fourier-transformed latent vector $\mathcal{F}(\mathbf{Z}_T)$ with the binary mask $\mathbf{M}$,

$$\mathcal{F}(\mathbf{Z}_T)[ic, :, :] = (1 - \mathbf{M}) \odot \mathcal{F}(\mathbf{Z}_T)[ic, :, :] + \mathbf{M} \odot \mathbf{W}, \tag{5}$$

where $ic$ is the watermark injecting channel and $\odot$ denotes the element-wise product. We denote the latent vector after watermarking as $\mathbf{Z}_T \oplus \mathbf{W}$.

**Latent Vector Optimization.** ZoDiac then seeks to find a latent vector $\mathbf{Z}_T$ such that, when being watermarked by $\mathbf{W}$ and then processed by the denoising process $\mathcal{G}$, generates an image $\hat{x}_0$ that is the most similar to the original image $x_0$. To solve the optimization problem, we design a reconstruction loss $\mathcal{L}$ that allows ZoDiac to iteratively refine $\mathbf{Z}_T$ via standard backpropagation:

$$\mathcal{L} = \mathcal{L}_2(x_0, \hat{x}_0) + \lambda_s \mathcal{L}_s(x_0, \hat{x}_0) + \lambda_p \mathcal{L}_p(x_0, \hat{x}_0), \tag{6}$$

where $\hat{x}_0 = \mathcal{G}(\mathbf{Z}_T \oplus \mathbf{W})$, $\mathcal{L}_2$ is the Euclidean distance, $\mathcal{L}_s$ represents the SSIM loss [50], $\mathcal{L}_p$ corresponds to the Watson-VGG perceptual loss [12], and $\lambda_s, \lambda_p$ are weighting coefficients. These coefficients are set to balance the scale of each loss component.

### 3.2.3 Adaptive Image Enhancement

Adaptive image enhancement aims to improve the visual quality of the image $\hat{x}_0$ generated from the watermark encoding step. To do this, it adaptively mixes $\hat{x}_0$ with the original image $x_0$ such that the *mixed image* can meet a desired image quality threshold. Mathematically, the mixed image $\bar{x}_0$ is:

$$\bar{x}_0 = \hat{x}_0 + \gamma(x_0 - \hat{x}_0), \tag{7}$$

where $\gamma \in [0, 1]$ is a modulating factor. A higher $\gamma$ improves image quality at the cost of potential watermark diminishing. Adaptive image enhancement automatically identifies the smallest $\gamma$ that results in desired image quality through binary search. It optimizes the following objective:

$$\min \gamma, \quad \text{s.t.} \quad S(\bar{x}_0, x_0) \geq s^*, \tag{8}$$

where $s^*$ is the desired image quality and $S$ is an image similarity metric. We use the SSIM metric by default.

## 3.3 Watermark Detection

In the watermark detection phase, our primary objective is to verify whether a given image $x_0$ contains watermark $\mathbf{W}$. The detection process starts with transforming $x_0$ to $\mathbf{y} = \mathcal{F}(\mathcal{G}'(x_0))[-1, :, :] \in \mathbb{C}^{w \times h}$, representing the last channel of the Fourier transformed latent vector. It then detects the presence of $\mathbf{W}$ in $\mathbf{y}$ via a statistical test procedure. The statistical test computes a $p$-value, an interpretable statistical metric that quantifies the likelihood of the observed watermark manifesting in a natural image by random chance. A watermark is considered detected when the computed $p$-value falls below a chosen threshold.

To compute the $p$-value, we first define the null hypothesis as $H_0 : \mathbf{y} \sim \mathcal{N}(\mathbf{0}, \sigma^2, \mathbf{I}_{\mathbb{C}})$, where $\sigma^2$ is estimated for each image from the variance of $\mathbf{y}$ masked by the circular binary mask $\mathbf{M}$, i.e., $\sigma^2 = \frac{1}{\sum \mathbf{M}} \sum (\mathbf{M} \odot \mathbf{y})^2$. This is because the DDIM inversion $\mathcal{G}'$ maps any test image $x_0$ into a Gaussian distribution [31], and the Fourier transformation of a Gaussian distribution remains Gaussian.

To test this hypothesis, we define a distance score $\eta$ that measures the disparity between $\mathbf{W}$ and $\mathbf{y}$ in the area defined by the binary mask $\mathbf{M}$:

$$\eta = \frac{1}{\sigma^2} \sum (\mathbf{M} \odot \mathbf{W} - \mathbf{M} \odot \mathbf{y})^2. \tag{9}$$

Under $H_0$, $\eta$ follows a non-central chi-squared distribution [26], characterized by $\sum \mathbf{M}$ degrees of freedom and a non-centrality parameter $\lambda = \frac{1}{\sigma^2} \sum (\mathbf{M} \odot \mathbf{W})^2$. An image is classified as watermarked if the value of $\eta$ is too small to occur by random chance. The probability of observing a value no larger than $\eta$, i.e., the $p$-value, is derived from the cumulative distribution function of the non-central chi-squared distribution. Non-watermarked images will exhibit higher $p$-values, while watermarked images yield lower values, indicating a successful rejection of $H_0$ and confirming the watermark's presence. In practice, we treat $(1 - p)$-value as the likelihood of watermark presence and set up a detection threshold $p^*$ to determine the watermark presence An image with $(1 - p) > p^*$ is considered to be watermarked.

# 4 Empirical Evaluation

This section evaluates ZoDiac's efficacy using a diverse domain of images including real photographs, AI-generated content, and visual artwork.

## 4.1 Experimental Setup

**Datasets.** Our evaluation uses images from three domains, photographs, AI-generated images, and visual artwork. For each domain, we randomly sample 500 images from well-established benchmarks, including **MS-COCO** [23], **DiffusionDB**, and **WikiArt**.

**ZoDiac Settings.** We use the pre-trained stable diffusion model *stable-diffusion-2-1-base* [29] with 50 denoising steps. We show ZoDiac is compatible with other diffusion models in § 4.4. We optimize the trainable latent vector for a maximum of 100 iterations using the Adam optimizer [20]. It takes

Table 1: Image quality in terms of PSNR, SSIM, and LPIPS, and watermark robustness in terms of Watermark Detection Rate (WDR) before and after attacks, on **MS-COCO**, **DiffusionDB**, and **WikiArt** datasets. We evaluate on ten individual attacks and two composite attacks: "All" that combines all the individual attacks, and "All w/o ↻" that excludes the rotation attack. ↑ and ↓ indicate whether higher or lower values are better. For each attack, we highlighted in gray the techniques with the maximum WDR, and those within 2% of the maximum; ZoDiac is the only method within 2% of the maximum WDR for all attacks, except Rotation. ZoDiac dominates all methods when facing the most advanced Zhao23 attack (recall Figure 1) and the combined All w/o ↻.

| Watermarking Method | Image Quality | | | Watermark Detection Rate (WDR) ↑ before and after Attacking | | | | | | | | | | | | | |
|---|---|---|---|---|---|---|---|---|---|---|---|---|---|---|---|---|---|
| | PSNR ↑ | SSIM ↑ | LPIPS ↓ | Pre-Attack | Post-Attack | | | | | | | | | | | | |
| | | | | | Brightness | Contrast | JPEG | G-Noise | G-Blur | BM3D | Bmshj18 | Cheng20 | Zhao23 | All w/o ↻ | Rotation | All | |
| DwtDct | 37.88 | 0.97 | 0.02 | 0.790 | 0.000 | 0.000 | 0.000 | 0.687 | 0.156 | 0.000 | 0.000 | 0.000 | 0.000 | 0.000 | 0.000 | 0.000 | **MS-COCO** |
| DwtDctSvd | 38.06 | 0.98 | 0.02 | 1.000 | 0.098 | 0.100 | 0.746 | 0.998 | 1.000 | 0.452 | 0.016 | 0.032 | 0.124 | 0.000 | 0.000 | 0.000 | |
| RivaGAN | 40.57 | 0.98 | 0.04 | 1.000 | 0.996 | 0.998 | 0.984 | 1.000 | 1.000 | 0.974 | 0.010 | 0.010 | 0.032 | 0.000 | 0.000 | 0.000 | |
| SSL | 41.81 | 0.98 | 0.06 | 1.000 | 0.992 | 0.996 | 0.046 | 0.038 | 1.000 | 0.000 | 0.000 | 0.000 | 0.000 | 0.000 | 0.952 | 0.000 | |
| CIN | 41.77 | 0.98 | 0.02 | 1.000 | 1.000 | 1.000 | 0.944 | 1.000 | 1.000 | 0.580 | 0.662 | 0.666 | 0.478 | 0.002 | 0.216 | 0.000 | |
| StegaStamp | 28.64 | 0.91 | 0.13 | 1.000 | 0.998 | 0.998 | 1.000 | 0.998 | 1.000 | 0.998 | 0.998 | 1.000 | 0.286 | 0.002 | 0.000 | 0.000 | |
| **ZoDiac** | 29.41 | 0.92 | 0.09 | 0.998 | 0.998 | 0.998 | 0.992 | 0.996 | 0.996 | 0.994 | 0.992 | 0.986 | 0.988 | 0.510 | 0.538 | 0.072 | |
| DwtDct | 37.77 | 0.96 | 0.02 | 0.690 | 0.000 | 0.000 | 0.000 | 0.574 | 0.224 | 0.000 | 0.000 | 0.000 | 0.000 | 0.000 | 0.000 | 0.000 | **DiffusionDB** |
| DwtDctSvd | 37.84 | 0.97 | 0.02 | 0.998 | 0.088 | 0.088 | 0.812 | 0.982 | 0.996 | 0.686 | 0.014 | 0.030 | 0.116 | 0.000 | 0.000 | 0.000 | |
| RivaGAN | 40.60 | 0.98 | 0.04 | 0.974 | 0.932 | 0.932 | 0.898 | 0.958 | 0.966 | 0.858 | 0.008 | 0.004 | 0.024 | 0.000 | 0.000 | 0.000 | |
| SSL | 41.84 | 0.98 | 0.06 | 0.998 | 0.990 | 0.996 | 0.040 | 0.030 | 1.000 | 0.000 | 0.000 | 0.000 | 0.000 | 0.000 | 0.898 | 0.000 | |
| CIN | 39.99 | 0.98 | 0.02 | 1.000 | 1.000 | 1.000 | 0.942 | 0.998 | 0.998 | 0.624 | 0.662 | 0.660 | 0.498 | 0.002 | 0.212 | 0.000 | |
| StegaStamp | 28.51 | 0.90 | 0.13 | 1.000 | 0.998 | 0.998 | 1.000 | 0.998 | 0.998 | 1.000 | 0.996 | 0.998 | 0.302 | 0.002 | 0.000 | 0.000 | |
| **ZoDiac** | 29.18 | 0.92 | 0.07 | 1.000 | 0.998 | 0.998 | 0.994 | 0.998 | 1.000 | 1.000 | 0.994 | 0.992 | 0.988 | 0.548 | 0.558 | 0.086 | |
| DwtDct | 38.84 | 0.97 | 0.02 | 0.754 | 0.000 | 0.000 | 0.000 | 0.594 | 0.280 | 0.000 | 0.000 | 0.000 | 0.000 | 0.000 | 0.000 | 0.000 | **WikiArt** |
| DwtDctSvd | 39.14 | 0.98 | 0.02 | 1.000 | 0.096 | 0.094 | 0.698 | 1.000 | 1.000 | 0.668 | 0.034 | 0.072 | 0.116 | 0.000 | 0.000 | 0.000 | |
| RivaGAN | 40.44 | 0.98 | 0.05 | 1.000 | 0.998 | 1.000 | 0.990 | 1.000 | 1.000 | 0.992 | 0.010 | 0.024 | 0.024 | 0.000 | 0.000 | 0.000 | |
| SSL | 41.81 | 0.99 | 0.06 | 1.000 | 0.988 | 0.988 | 0.082 | 0.108 | 1.000 | 0.000 | 0.000 | 0.000 | 0.000 | 0.000 | 0.932 | 0.000 | |
| CIN | 41.92 | 0.98 | 0.02 | 1.000 | 1.000 | 1.000 | 0.936 | 0.998 | 1.000 | 0.572 | 0.664 | 0.664 | 0.436 | 0.002 | 0.228 | 0.000 | |
| StegaStamp | 28.45 | 0.91 | 0.14 | 1.000 | 1.000 | 0.998 | 1.000 | 1.000 | 1.000 | 1.000 | 1.000 | 1.000 | 0.182 | 0.002 | 0.000 | 0.000 | |
| **ZoDiac** | 30.04 | 0.92 | 0.10 | 1.000 | 1.000 | 1.000 | 1.000 | 0.998 | 1.000 | 1.000 | 0.994 | 0.994 | 0.992 | 0.530 | 0.478 | 0.082 | |

45.4 – 255.9s to watermark one image (details see Appendix D). We calibrate the weights for the SSIM loss $\lambda_s$ and the perceptual loss $\lambda_p$ to 0.1 and 0.01, respectively, to balance the scales of the various loss components. We set the watermark injecting channel $ic$ to be the last channel of the latent representation and the watermark radius $d^*$ to 10. To balance the image quality and detectability of the watermark, we set the SSIM threshold $s^*$ to 0.92 and the detection threshold $p^*$ to 0.9, except where explicitly noted otherwise. We evaluate other hyperparameter settings in the ablation study (see §4.4 and Appendix B).

**Watermarking Baselines.** We compare ZoDiac to six watermarking methods.

- **Traditional Methods:** Two conventional techniques, **DwtDct** and **DwtDctSvd** [9], utilize frequency decomposition. DwtDctSvd has been used to watermark stable diffusion models [29].
- **Training-Based Methods: RivaGAN** [48] is a pre-trained GAN-based model incorporating an attention mechanism. **SSL** [16] is a latent-space–based watermarking method that uses self-supervised learning for network pre-training. **CIN** [25] combines invertible and non-invertible mechanisms to achieve high imperceptibility and robustness against strong noise attacks. **StegaStamp** [35] applies adversarial training and integrates both CNN and spatial transformer techniques.

Considering the above methods inject bitstrings as the watermark, we use 32-bit messages for DwtDct, DwtDctSvd, RivaGAN, SSL, and CIN, and 96-bit messages for StegaStamp. Detection thresholds were set to reject the null hypothesis (i.e., $H_0$: the image has no watermark) with $p < 0.01$, requiring correct detection of $24/32$ and $61/96$ bits for the respective methods to form a fair comparison [51].

**Watermark Attack Methods.** We evaluate ZoDiac's robustness against a comprehensive set of attacks used in recent watermarking evaluations [51, 1]. The set of attacks employed includes:

- Adjustments in **brightness** or **contrast** with a factor of 0.5.
- **JPEG compression** with a quality setting of 50.
- **Image rotation** by 90 degrees.
- Addition of **Gaussian noise** with a std of 0.05.
- **Gaussian blur** with a kernel size of 5 and std of 1.
- **BM3D denoising** algorithm with a std of 0.1.
- **VAE-based image compression** models, *Bmshj18* [2] and *Cheng20* [8], with a quality level of 3.
- A **stable diffusion-based image regeneration** model, *Zhao23* [51] with 60 denoising steps.

Since the attack strength tuned by the hyper-parameters of different attack methods will influence the presence and detection of the injected watermark, we evaluate the effectiveness of ZoDiac under other settings in the ablation study (see §4.4 and Appendix B). Besides, unlike prior work, we go deeper and evaluate against the composite attack that combines all the aforementioned attacks (named "All") and a variant without rotation (named "All w/o ↻").

We also examine the robustness of ZoDiac under *pipeline-aware attack*, where the attacker has partial or full knowledge of our watermarking method, in Appendix C.2. Our results confirm the robustness of ZoDiac in the face of partial knowledge attacks and underscore the critical importance of safeguarding the model weights and watermark configurations.

**Evaluation Metrics.** We evaluate the quality of the watermarked image ($\bar{x}$) compared to the original image ($x$) using three metrics: Peak Signal-to-Noise Ratio $(\text{PSNR})(x, \bar{x}) = -10 \log_{10}(\text{MSE}(x, \bar{x}))$, Structural Similarity Index (SSIM) [40] with range $[0, 1]$, and LPIPS [49] that measures perceptual similarity. For PSNR and SSIM, larger values correspond to higher image similarity; for LPIPS, lower values do. To assess watermark robustness, we use the *Watermark Detection Rate (WDR)* for watermarked images, which is the same as True Positive Rate (TPR), and *False Positive Rate (FPR)* for non-watermarked images. Both WDR and FPR range from 0 to 1. We expect to achieve high WDR and low FPR.

## 4.2 Results

Table 1 reports the watermarked image quality and WDR before and after attacks for ZoDiac and baselines.

**Image Quality.** ZoDiac achieves satisfactory image perceptual similarity, as defined by [7] and [34], of approximately 30dB PSNR and 0.92 SSIM. ZoDiac's image quality outperforms the previously most robust watermarking method, StegaStamp. Figure 11 in Appendix E illustrates the visually imperceptible effect of ZoDiac's and other methods' watermarks. §4.4 examines how increasing ZoDiac's SSIM threshold can further improve image quality, trading off WDR.

**Watermark Robustness.** ZoDiac consistently exhibits high detection rates, within $2\%$ of the maximum WDR, against watermark removal algorithms as highlighted by the gray cells (except for the *Rotation* attack). Traditional approaches, such as DwtDct and DwtDctSvd, consistently fail with brightness and contrast changes, rotation, and advanced generated-AI-based attacks. RivaGAN and SSL fail for advanced attacks, *Bmshj18*, *Cheng20*, and *Zhao23*. StegaStamp, despite being robust across most attacks, fails for rotation and demonstrates significantly diminished detection rates under the diffusion model-based attack *Zhao23*. When facing the composite attack excluding rotation ("All w/o ↻" column), ZoDiac maintains a detection rate of about $0.5$, while all baselines fail, with a WDR close to 0. Appendix B.8 provides more discussions on composite attacks.

The exception of rotation is noteworthy. All methods, apart from SSL which incorporates rotation during training, fail for rotational disturbances. In ZoDiac, this limitation stems from the non-rotation-invariant nature of the DDIM inversion process; the latent representation derived from an image and its rotated version differ significantly. However, rotation is not an invisible attack and its effects are readily reversible, allowing users to manually correct image orientation to facilitate watermark detection. Appendix C.1 presents an extended discussion on overcoming rotational attacks.

## 4.3 Why ZoDiac is Attack Resilient

Recall that the image generation process is an iterative algorithm that progresses from $x_T$ to $x_0$, involving the prediction of an approximate $x_0$ at each step and the subsequent addition of noise to obtain the noisy input for the next step. We hypothesize that this reciprocating denoising process may inherently enhance the robustness of the watermark when faced with strong attacks. To validate the hypothesis, we experiment with different denoising steps when reconstructing the watermarked image on the MS-COCO dataset. In addition to the commonly-used 50 denoising steps, we further evaluate denoising steps of 10, 1, and even 0, where 0 means utilizing only the well-trained image autoencoder in the diffusion model without the diffusion process.

Table 2 reports the watermarked image quality in terms of PSNR when setting the SSIM threshold $s^* = 0.92$ and the robustness when subjected to three representative attacks. Appendix B.2 contains the full table with all attacks. We make two observations. First, the number of denoising steps

Table 2: The effects of varying denoising steps on image quality (PSNR) and watermark detection rate (WDR). The denoising step of 0 means utilizing only the image autoencoder in the diffusion model without the diffusion process.

| Denoising Steps | PSNR↑ | WDR↑ before and after attack | | | |
| --- | --- | --- | --- | --- | --- |
| | | Pre-Attack | Post-Attack | | |
| | | | Rotation | Zhao23 | All w/o ↻ |
| 50 | 29.41 | 0.998 | 0.538 | 0.988 | 0.510 |
| 10 | 29.51 | 0.996 | 0.534 | 0.976 | 0.502 |
| 1 | 28.67 | 0.992 | 0.532 | 0.97 | 0.498 |
| 0 | 29.09 | 0.942 | 0.226 | 0.712 | 0.002 |

Table 3: The effects of varying detection thresholds $p^* \in \{0.90, 0.95, 0.99\}$ on watermark detection rate (WDR) and false positive rate (FPR). ↑ and ↓ indicate whether higher or lower values are better.

| Detection Threshold $p^*$ | FPR↓ | WDR↑ before and after attack | | | |
| --- | --- | --- | --- | --- | --- |
| | | Pre-Attack | Post-Attack | | |
| | | | Rotation | Zhao23 | All w/o ↻ |
| 0.90 | 0.062 | **0.998** | **0.538** | **0.988** | **0.510** |
| 0.95 | 0.030 | 0.998 | 0.376 | 0.974 | 0.372 |
| 0.99 | **0.004** | 0.992 | 0.106 | 0.938 | 0.166 |

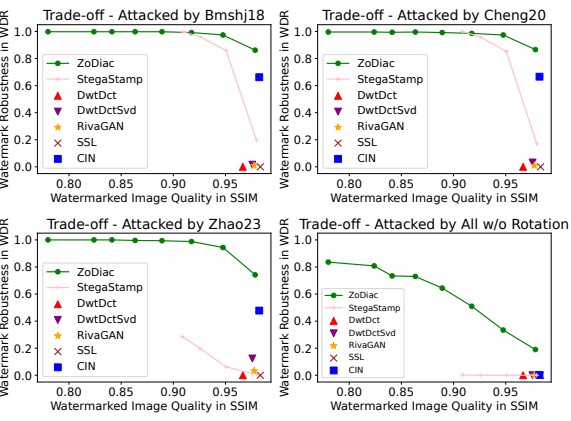

Figure 3: The trade-off between the watermarked image quality (SSIM) and the watermark detection rate (WDR) on **MS-COCO** dataset. The image quality is controlled by SSIM threshold $s^* \in [0.8, 0.98]$ in increments of 0.03 and the robustness is evaluated post-attack with four advanced attack methods.

Figure 4: ZoDiac exhibits comparable results when using different pre-trained stable diffusion models, *stable-diffusion-2-1-base*, *stable-diffusion-v1-4*, and *stable-diffusion-xl-base-1.0*. Colors represent attacks; "None" represents no attack.

do not significantly influence the watermarking performance. Regardless of the attack methods employed, ZoDiac exhibits consistently high WDR while maintaining similar image quality with different denoising steps. Second, even when reduced to a single denoising step, ZoDiac significantly outperforms the variant without the diffusion process (i.e., when the denoising step is 0) in terms of watermark robustness, thereby verifying the hypothesis that the diffusion model contributes substantially to the robustness of the embedded watermark.

## 4.4 Ablation Study

**Varying the SSIM Threshold $s^*$.** While a high image quality threshold $s^*$ can yield an image with negligible quality loss, it conversely raises the risk of watermark elimination. This study explores the impact of varying $s^*$ on image quality and watermark robustness. Figure 3 illustrates the trade-off curves on the MS-COCO dataset, subject to four advanced attack methods, by adjusting SSIM thresholds $s^* \in [0.8, 0.98]$ in increments of 0.03. Appendix B.3 reports results for eight other attacks and two other datasets. Overall, *against advanced watermark removal techniques, ZoDiac consistently outperforms all baseline methods in terms of robustness.* Although the adaptive image enhancement step can also enhance the watermarked image quality of StegaStamp, the most robust baseline method, it causes a significant decline in its watermark robustness, as shown by the pink line in Figure 3. StegaStamp injects watermarks in the image space, making it more sensitive to the enhancement. By contrast, ZoDiac injects watermarks in the latent space of images, benefiting from the enhancement step and achieving a balance between high image quality and watermark robustness.

**Varying the Detection Threshold $p^*$.** Recall from §3.3 that an image is considered watermarked if its $(1-p)$-value exceeds $p^*$. A larger $p^*$ imposes a stricter criterion for watermark detection, reducing the

Table 4: The PSNR of attacked watermarked images compared to those without being attacked and the WDR of ZoDiac and StegaStamp under different strengths from 0.2 to 1.0 (i.e., no attack) of the **Brightness** attack (left) and the **Contrast** (right) attack.

| | Detection Threshold | FPR | WDR under Brightness Factor of | | | | | | Detection Threshold | FPR | WDR under Contrast Factor of | | | | |
|---|---|---|---|---|---|---|---|---|---|---|---|---|---|---|---|
| | | | 0.2 | 0.4 | 0.6 | 0.8 | 1.0 | | | | 0.2 | 0.4 | 0.6 | 0.8 | 1.0 |
| PSNR | - | - | 11.41 | 14.56 | 27.95 | 33.58 | 100.0 | PSNR | - | - | 14.54 | 23.03 | 30.56 | 36.57 | 100.0 |
| StegaStamp | 61/96 | 0.056 | 0.852 | 0.988 | 1.0 | 1.0 | 1.0 | StegaStamp | 61/96 | 0.056 | 0.730 | 0.98 | 1.0 | 1.0 | 1.0 |
| StegaStamp | 60/96 | 0.094 | 0.890 | 1.0 | 1.0 | 1.0 | 1.0 | StegaStamp | 60/96 | 0.094 | 0.768 | 0.998 | 1.0 | 1.0 | 1.0 |
| ZoDiac | $p^* = 0.95$ | 0.032 | 0.994 | 0.994 | 0.998 | 0.998 | 0.998 | ZoDiac | $p^* = 0.95$ | 0.032 | 0.998 | 0.996 | 0.998 | 0.998 | 0.998 |
| ZoDiac | $p^* = 0.90$ | 0.062 | 0.996 | 0.998 | 0.998 | 0.998 | 0.998 | ZoDiac | $p^* = 0.90$ | 0.062 | 0.994 | 0.998 | 0.998 | 0.998 | 0.998 |

probability of both watermark detection for watermarked images, measured by watermark detection rate (WDR), and false detection for non-watermarked images, measured by false positive rate (FPR). Table 3 reports the effects of detection thresholds $p^* \in \{0.90, 0.95, 0.99\}$ on WDR and FPR before and after three representative attacks on MS-COCO dataset. Appendix B.4 contains the full table with all attacks for the three datasets and also provides the FPR of baselines for reference. *The data show that $p^* = 0.9$ maintains a high WDR and an acceptable FPR, in practice.*

**Varying the Backbone Models.** ZoDiac is compatible with different pre-trained stable diffusion models. In this study, we extend our evaluation to the *stable-diffusion-v1-4* and *stable-diffusion-xl-base-1.0* (in addition to *stable-diffusion-2-1-base*). Figure 4 shows the trade-off curves between the watermarked image quality and the watermark detection rate without attack as well as with three representative attacks, *Rotation*, *Zhao23*, and *All w/o ↻*. Appendix B.5 shows additional results addressing other attacks. The data shows consistent performance (WDR and image quality in SSIM) between the two models, as indicated by the almost overlapped solid and dashed lines, suggesting that ZoDiac can seamlessly integrate with other pre-trained stable diffusion models, maintaining its state-of-the-art efficiency, regardless of the backbone model employed.

**Varying the Attack Strength.** We also compare ZoDiac with the strongest baseline StegaStamp under different hyper-parameter settings of the attack methods, in terms of WDR under different FPR levels. Table 4 reports the image quality in terms of PSNR and the watermarking robustness when we adjust the *Brightness* and *Contrast* attacks with a factor of 0.2 to 1.0 following WAVES [1]. As the adjustment factors for brightness and contrast decrease from 0.9 to 0.2, the PSNR of the attacked watermarked images drops significantly—from 37.79 to 11.41 for brightness and from 38.57 to 14.54 for contrast. Despite this degradation, ZoDiac consistently maintains a WDR above 0.99 across all scenarios. Even at extreme settings, where PSNRs fall to 11.41 for brightness and 14.54 for contrast, ZoDiac significantly outperforms StegaStamp, achieving higher WDR and lower FPR. *This clearly demonstrates ZoDiac 's robustness even under severe quality degradation.* Appendix B.6 includes additional results when tuning other attacks.

## 5 Contributions

We have presented ZoDiac, a robust watermarking framework based on pre-trained stable diffusion models. ZoDiac hides watermarks within the trainable latent space of a pre-trained stable diffusion model, enhancing watermark robustness while preserving the perceptual quality of the watermarked images. An extensive evaluation across three datasets of photographs, AI-generated images, and art demonstrates ZoDiac's effectiveness in achieving both invisibility and robustness to an array of watermark removal attacks. In particular, ZoDiac is robust to advanced generative-AI-based removal methods and composite attacks, while prior watermarking methods fail in these scenarios. This robustness, coupled with the ability to achieve high-quality watermarked images, positions ZoDiac as a significant advancement in the field of image watermarking. ZoDiac focuses on zero-bit watermarking [17], only hiding and detecting a mark. ZoDiac is constrained to adhere to a Gaussian distribution in the latent vector of the diffusion model, but future work will explore encoding meaningful information, such as a message, in the watermark while preserving ZoDiac's robustness.

## Acknowledgments and Disclosure of Funding

This work is supported in part by the National Science Foundation under grants no. CCF-2210243, DMS-2220211, CNS-2224054, CNS-2338512, and CNS-2312396.

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

# A    Additional Related Work

**Digital Image Watermarking.** Digital watermarking, particularly in the image domain, has been a cornerstone in computer vision for decades. Traditional methods have predominantly relied on frequency decomposition techniques such as Discrete Fourier Transform (DFT) [37], Discrete Cosine Transform (DCT) [6], Discrete Wavelet Transform (DWT) [44], or their combinations [9]. These frequency-based methods are advantageous due to their inherent resilience to common image manipulations.

The development of deep neural networks (DNN) has introduced novel learning-based watermarking approaches. HiDDeN [53] employs joint training of watermark encoder and decoder alongside noise layers that mimic image perturbations. It leverages an adversarial discriminator for enhanced visual quality and has been expanded for arbitrary image resolutions and message lengths in [22]. Regarding the watermark robustness, StegaStamp [35] uses differentiable image perturbations during training to improve noise resistance and incorporates a spatial transformer network for minor perspective and geometric changes. Distortion Agnostic [24] further brings robustness to unforeseen transformations by introducing adversarial training. Considering the watermarking as an image generation process, RivaGAN [48], SteganoGAN [47], and ARWGAN [19] resort to generative AI, particularly Generative Adversarial Network (GAN), for enhanced performance and robustness in watermarking. A comprehensive review is provided in [38]. With the demonstrated efficacy of diffusion models in image generation [14], our work pioneers the use of pre-trained diffusion models for robust image watermarking.

**Image Watermarking Attack.** Watermarking attacks are typically categorized into two types [51]. Destructive attacks treat the watermark as part of the image, seeking to remove it through image corruption. Common methods include altering image brightness or contrast, applying JPEG compression, introducing Gaussian noise, and so on. Constructive attacks, on the other hand, view the watermark as noise superimposed on the original image, focusing on its removal via image purification techniques such as Gaussian blur [18], BM3D [13], and learning-based methods like DnCNNs [46]. Recently, regeneration attacks have emerged, leveraging the strengths of both destructive and constructive approaches. These attacks corrupt the image by adding Gaussian noise to its latent representation, then reconstruct it using generative models such as Variational AutoEncoders [2, 8] or Stable Diffusion [29]. We aim to effectively counter watermark removal attacks especially advanced generative AI-based ones when watermarking given images.

**Diffusion Model Watermarking.** Rapid evolution in deep generative models has led to methods capable of synthesizing high-quality, realistic images [14]. These models pose threats to society due to their potential misuse, prompting the necessity to differentiate AI-generated images from those created by humans. Recent efforts resort to watermarking the model itself, which focuses on enabling these models to automatically generate watermarked images for easier identification later. One straightforward approach is to fine-tune diffusion models with datasets containing watermarked images, leading these models to inherently produce watermarked outputs [11, 39, 52, 45]. Alternatives like Stable Signature [15] focus on training a dedicated watermark decoder to fine-tune only the latent decoder part of the stable diffusion model. Tree-Rings [42] takes a different route by embedding watermarks in the initial noisy latent, detectable through DDIM inversion [14]. Although these methods effectively lead diffusion models to generate watermarked images, they are not equipped for watermarking existing images, a functionality our approach is designed to provide.

# B    Additional Ablation Study

## B.1    Effectiveness of Latent Vector Initialization

Figure 5 compares the synthetic images at different optimization iterations when training the latent vector initialized with the vector from DDIM inversion (a) and a vector randomly sampled from the Gaussian distribution (b). The result shows the advantage of the latent vector initialization step, which achieves a higher quality image, measured by the $\mathcal{L}_2$ distance between the image and its original counterpart within only 20 iterations. In contrast, a randomly initialized latent vector requires more than hundreds of iterations to produce a similar image whose quality ($\mathcal{L}_2 = 0.0036$) still largely falls behind.

## B.2    Why ZoDiac is Attack Resilient Continued

Table 5 completes Table 2 from §4.3, showing the effects of different denoising steps when reconstructing the wa-

Input Image    $it = 0, \mathcal{L}_2 = 0.0056$    $it = 5, \mathcal{L}_2 = 0.0020$    $it = 20, \mathcal{L}_2 = 0.0015$

(a) Images generated from prepared initial latent with DDIM inversion

$it = 0, \mathcal{L}_2 = 1109$    $it = 20, \mathcal{L}_2 = 0.0376$    $it = 200, \mathcal{L}_2 = 0.0117$    $it = 500 \ \mathcal{L}_2 = 0.0036$

(b) Images generated from random initial latent

Figure 5: Synthetic images along with the optimization iterations and $\mathcal{L}_2$ loss values. We observe much faster convergence when (a) using the latent vector initialized from the DDIM inversion compared to (b) using a randomly initialized latent vector.

termarked images on WDR under all attacks. We measure WDR on watermarked images generated with the SSIM threshold $s^* = 0.92$. Overall, ZoDiac exhibits consistently high WDR with different denoising steps,

Table 5: The effects of varying denoising steps on image quality and watermark detection rate (WDR). The denoising step of 0 means utilizing only the image autoencoder in the diffusion model without the diffusion process. ↑ and ↓ indicate whether higher or lower values are better.

| Denoising Steps | Image Quality | | | Pre-Attack | Watermark Detection Rate (WDR) ↑ before and after attack | | | | | | | | | | | |
|---|---|---|---|---|---|---|---|---|---|---|---|---|---|---|---|---|
| | | | | | Post-Attack | | | | | | | | | | | |
| | PSNR ↑ | SSIM ↑ | LPIPS ↓ | | Brightness | Contrast | JPEG | G-Noise | G-Blur | BM3D | Bmshj18 | Cheng20 | Zhao23 | All w/o ↻ | Rotation | All |
| 50 | 29.41 | 0.92 | 0.09 | 0.998 | 0.998 | 0.998 | 0.992 | 0.996 | 0.996 | 0.994 | 0.992 | 0.986 | 0.988 | 0.510 | 0.538 | 0.072 |
| 30 | 29.44 | 0.92 | 0.08 | 0.996 | 0.998 | 0.996 | 0.992 | 0.99 | 0.996 | 0.994 | 0.99 | 0.986 | 0.982 | 0.510 | 0.538 | 0.07 |
| 10 | 29.51 | 0.92 | 0.08 | 0.996 | 0.998 | 0.994 | 0.992 | 0.990 | 0.996 | 0.994 | 0.986 | 0.984 | 0.976 | 0.502 | 0.534 | 0.068 |
| 1 | 28.67 | 0.92 | 0.10 | 0.992 | 0.99 | 0.986 | 0.988 | 0.988 | 0.992 | 0.994 | 0.98 | 0.978 | 0.97 | 0.498 | 0.532 | 0.060 |
| 0 | 29.09 | 0.92 | 0.08 | 0.942 | 0.674 | 0.938 | 0.888 | 0.9 | 0.838 | 0.744 | 0.738 | 0.7 | 0.712 | 0.002 | 0.226 | 0.000 |

demonstrating that different denoising steps do not influence the watermark robustness. In the meanwhile, when the denoising step becomes 0, the robustness drops significantly, which verifies the hypothesis that the diffusion process contributes substantially to the robustness of the embedded watermark.

## B.3 Varying the SSIM Threshold $s^*$ Continued

Recall from §3.2.3 that the final watermarked image $\bar{x}_0$ is derived by blending the raw watermarked image $\hat{x}_0$ with the original image $x_0$ to meet the desired image quality, specified by an SSIM threshold $s^*$. While setting a high threshold for image quality $s^*$ can result in minimal loss of image quality, it simultaneously decreases the likelihood of accurate watermark detection. Building on Figure 3 from §4.4, we explore the impact of varying $s^*$ values on the trade-off between image quality and watermark robustness on the full set of attacks and datasets. Figures 6, 7, and 8 show the trade-off curves on the MS-COCO, DiffusionDB, and WikiArt datasets, respectively. The x-axes show the watermarked image quality in SSIM and the y-axis shows the watermark detection rate, subject to twelve various attack scenarios. ZoDiac's image quality is controlled by SSIM thresholds $s^* \in [0.8, 0.98]$ in increments of 0.03.

§4.4 summarized two main observations, which are confirmed by these further experiments: First, ZoDiac produces watermark robustness that is on par with or surpasses that of the existing methods with similar image quality. For instance, when brightness or contrast changes, ZoDiac's WDR approximates that of RivaGAN and SSL. Notably, against more advanced watermark removal techniques, as depicted in the final row of Figures 6, 7, and 8, ZoDiac consistently outperforms all baseline methods in terms of robustness. Second, although adaptive image enhancement can also enhance the watermarked image quality of StegaStamp, the most robust baseline method, this approach markedly reduces its watermark robustness, as indicated by the pink lines in Figures 6, 7, and 8. This decrease in robustness occurs because StegaStamp embeds watermarks directly in the image space, rendering it more vulnerable to the proposed image quality enhancements. By contrast, ZoDiac injects watermarks in the latent space of images, allowing it to benefit from the enhancement step in achieving a balance between good image quality and high watermark robustness.

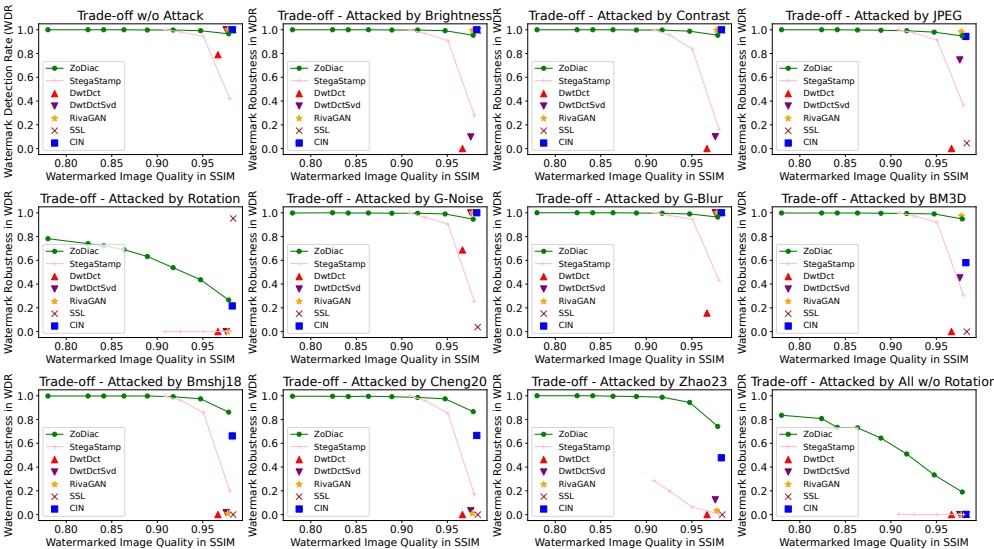

Figure 6: The trade-off between the watermarked image quality in terms of SSIM and the watermark robustness in terms of watermark detection rate (WDR) on **MS-COCO** dataset. The image quality is controlled by SSIM threshold $s^* \in [0.8, 0.98]$ with a step size of $0.03$ as described in §3.2.3 and the robustness is evaluated pre-attack and post-attack with different attack methods.

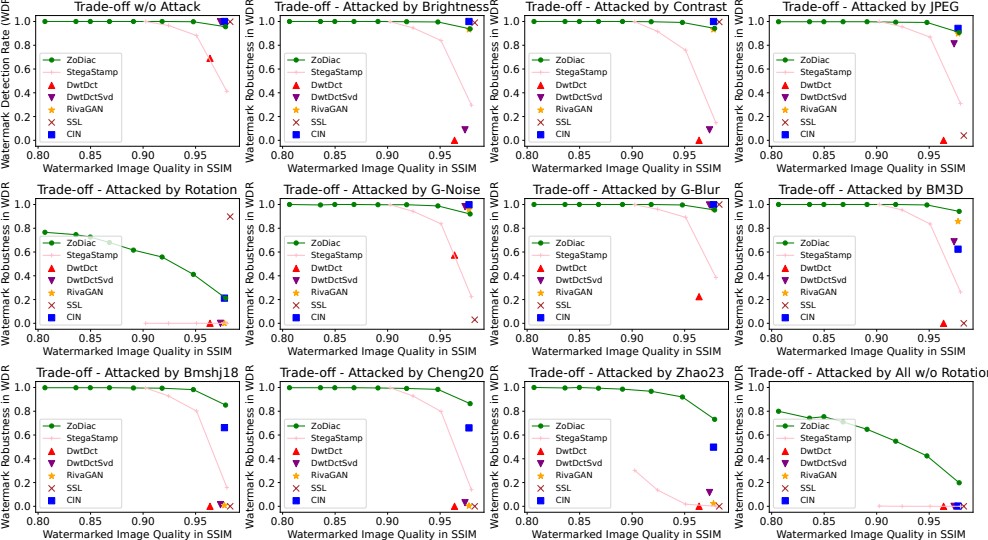

Figure 7: The trade-off between the watermarked image quality and the watermark robustness on **DiffusionDB** dataset similar to Figure 6.

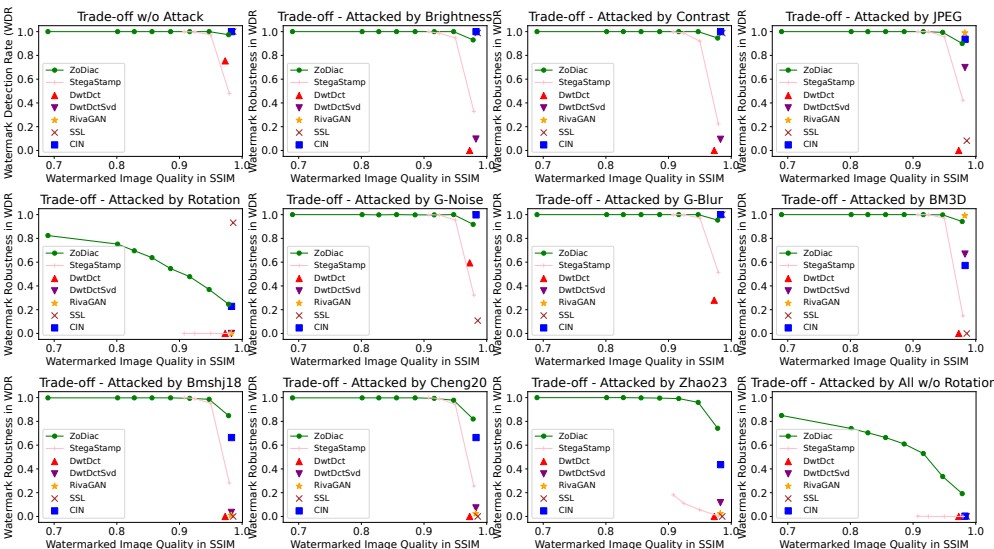

Figure 8: The trade-off between the watermarked image quality and the watermark robustness on **WikiArt** dataset similar to Figure 6.

## B.4 Varying the Detection Threshold p* Continued

Table 7 completes Table 3 from §4.4, showing the effects of detection thresholds $p^* \in \{0.90, 0.95, 0.99\}$ on WDR and FPR, under all attacks. We measure WDR on watermarked images generated with the SSIM threshold $s^* = 0.92$. The maximum WDR and WDR within 2% of the maximum are highlighted in gray. As expected, a higher $p^*$ usually corresponds to lower rates of both WDR and FPR. Specifically, under most attack conditions except *Rotation* and *All w/o ↻*, WDR exhibits a marginal decline when $p^*$ is increased. We further provide the FPR of existing methods on the MS-COCO dataset in Table 6 as a reference. When investigating the results in Table 1 in the main paper together with Table 6 and Table 7, we conclude that, under the same FPR level, ZoDiac always achieves a higher Watermark Detection Rate (WDR), demonstrating its superiority on watermark robustness.

Table 6: The FPR for existing methods on the MS-COCO dataset.

| Method | DwtDct | DwtDctSvd | RivaGAN | SSL | CIN | StegaStamp |
|--------|--------|-----------|---------|-----|-----|------------|
| FPR | 0.052 | 0.018 | 0.036 | 0 | 0.026 | 0.056 |

Table 7: The effects of varying detection thresholds $p^* \in \{0.90, 0.95, 0.99\}$ on watermark detection rate (WDR) and false positive rate (FPR) for all attacks. We measure WDR on watermarked images generated with SSIM threshold $s^* = 0.92$. We highlighted in gray the techniques with the maximum WDR, and those within 2% of the maximum. The data indicate that a higher $p^*$ usually corresponds to slightly lower WDR.

| Detection Threshold | FPR ↓ | Watermark Detection Rate (WDR) ↑ before and after attack | | | | | | | | | | | | |
| | | Pre-Attack | Post-Attack | | | | | | | | | | | |
| | | Brightness | Contrast | JPEG | G-Noise | G-Blur | BM3D | Bmshj18 | Cheng20 | Zhao23 | All w/o ↻ | Rotation | All |
|---|---|---|---|---|---|---|---|---|---|---|---|---|---|---|
| **MS-COCO** | | | | | | | | | | | | | | |
| 0.90 | 0.062 | 0.998 | 0.998 | 0.998 | 0.992 | 0.996 | 0.996 | 0.994 | 0.992 | 0.986 | 0.988 | 0.510 | 0.538 | 0.072 |
| 0.95 | 0.030 | 0.998 | 0.996 | 0.998 | 0.992 | 0.996 | 0.996 | 0.994 | 0.986 | 0.978 | 0.974 | 0.372 | 0.376 | 0.028 |
| 0.99 | 0.004 | 0.992 | 0.990 | 0.990 | 0.978 | 0.984 | 0.988 | 0.988 | 0.960 | 0.954 | 0.938 | 0.166 | 0.106 | 0.000 |
| **DiffusionDB** | | | | | | | | | | | | | | |
| 0.90 | 0.050 | 1.000 | 0.998 | 0.998 | 0.994 | 0.998 | 1.000 | 1.000 | 0.994 | 0.992 | 0.988 | 0.548 | 0.558 | 0.086 |
| 0.95 | 0.018 | 1.000 | 0.998 | 0.996 | 0.994 | 0.994 | 1.000 | 1.000 | 0.992 | 0.988 | 0.952 | 0.418 | 0.356 | 0.028 |
| 0.99 | 0.004 | 0.998 | 0.992 | 0.99 | 0.982 | 0.990 | 1.000 | 0.994 | 0.974 | 0.984 | 0.902 | 0.174 | 0.130 | 0.006 |
| **WikiArt** | | | | | | | | | | | | | | |
| 0.90 | 0.064 | 1.000 | 1.000 | 1.000 | 1.000 | 0.998 | 1.000 | 1.000 | 0.994 | 0.994 | 0.992 | 0.530 | 0.478 | 0.082 |
| 0.95 | 0.024 | 1.000 | 1.000 | 1.000 | 0.998 | 0.996 | 1.000 | 0.998 | 0.994 | 0.990 | 0.980 | 0.392 | 0.330 | 0.032 |
| 0.99 | 0.004 | 1.000 | 1.000 | 1.000 | 0.992 | 0.994 | 1.000 | 0.998 | 0.980 | 0.964 | 0.944 | 0.192 | 0.104 | 0.002 |

## B.5 Varying the Backbone Models Continued

Figure 9 completes Figure 4 from §4.4, which includes the trade-off curves between image quality and watermark robustness under the ten individual attacks and the two composite attacks when using different backbone stable diffusion models, *stable-diffusion-2-1-base*, *stable-diffusion-v1-4*, and *stable-diffusion-xl-base-1.0*. The figure

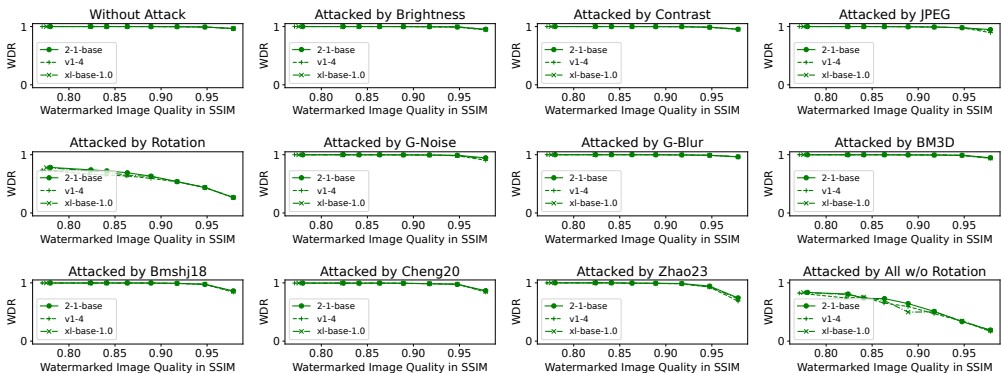

Figure 9: The watermarking performance comparisons between three pre-trained stable diffusion models, *stable-diffusion-2-1-base*, *stable-diffusion-v1-4*, and *stable-diffusion-xl-base-1.0*. The trade-off curves between the watermarked image quality and the watermark detection rate with and without attack are provided.

shows the trade-off curves between the watermarked image quality and the watermark detection rate with and without attack. Overall, the three backbone models exhibit similar watermarking performance.

## B.6 Varying the Attack Strength Continued

We conducted additional experiments under various attack levels, following the settings in WAVES [1]. The experiments span multiple attack scenarios:

- Tables 8 and 9: Adjustments in brightness or contrast with a factor of 0.2 to 1.0 (i.e., the watermarked images without being attacked).
- Table 10: JPEG compression with a quality setting from 10 to 90
- Table 11: Gaussian noise with a standard deviation varying from 0.02 to 0.1
- Table 12: Gaussian blur with a kernel size of 5 to 19.
- Tables 13 and 14: Two VAE-based image compression models, Bmshj18 and Cheng20 with quality levels of 2 to 6.
- Table 15: A stable diffusion-based image regeneration model, Zhao23 with 40 to 200 denoising steps.

We compared ZoDiac and the strongest baseline, StegaStamp, in terms of WDR under different FPR levels, leading to several insights:

**Brightness and Contrast Attacks**: As the adjustment factors for brightness and contrast decrease from 0.9 to 0.2, the PSNR of the attacked watermarked images drops significantly—from 37.79 to 11.41 for brightness and from 38.57 to 14.54 for contrast. **Despite this degradation, ZoDiac consistently maintains a WDR above 0.99 across all scenarios.** Even at extreme settings, where PSNRs fall to 11.41 for brightness and 14.54 for contrast, ZoDiac significantly outperforms StegaStamp, achieving higher WDR and lower FPR. Specifically, ZoDiac records a WDR of 0.994 at an FPR of 0.032 compared to StegaStamp's 0.852 at an FPR of 0.056 for brightness, and a WDR of 0.998 at an FPR of 0.032 compared to StegaStamp's 0.730 at an FPR of 0.056 for contrast. This clearly demonstrates ZoDiac 's robustness even under severe quality degradation.

**JPEG, G-Noise, G-Blur, Bmshj18, and Cheng20 Attacks**: With varying attack parameters, including the quality settings for JPEG, Bmshj18, Cheng20, the standard deviation for G-Noise the kernel size for G-Blur, the PSNR of the attacked watermarked images remains within the range of 20 to 40. **In these scenarios, both ZoDiac and StegaStamp consistently sustain a high WDR of around 0.98.** The only exception is JPEG with a quality setting of 10. Under this condition, the WDR of both ZoDiac and StegaStamp slightly decreases from their usual range of 0.956-0.986 to 0.746-0.834 for ZoDiac, and from 0.996-1.0 to 0.758-0.806 for StegaStamp.

**Zhao23**: As the number of image regeneration steps in Zhao23 increases, the PSNR of the attacked watermarked images decreases from 27.15 to 22.44. **Despite this degradation, ZoDiac remains highly resistant across most scenarios**, with its WDR slightly decreasing from 0.998 to 0.898. In contrast, StegaStamp significantly underperforms, completely failing when the regeneration steps exceed 120.

Table 8: The PSNR of attacked watermarked images compared to those without being attacked and the WDR of ZoDiac and StegaStamp under different strengths from 0.2 to 1.0 (i.e., no attack) of the **Brightness** attack.

| | Detection Threshold | FPR | WDR under Brightness adjustment factor of | | | | | | | | |
|---|---|---|---|---|---|---|---|---|---|---|---|
| | | | 0.2 | 0.3 | 0.4 | 0.5 | 0.6 | 0.7 | 0.8 | 0.9 | 1.0 |
| PSNR | - | - | 11.41 | 13.21 | 14.56 | 21.89 | 27.95 | 30.84 | 33.58 | 37.79 | 100.0 |
| StegaStamp | 61/96 | 0.056 | 0.852 | 0.954 | 0.988 | 0.998 | 1.0 | 1.0 | 1.0 | 1.0 | 1.0 |
| StegaStamp | 60/96 | 0.094 | 0.890 | 0.990 | 1.0 | 1.0 | 1.0 | 1.0 | 1.0 | 1.0 | 1.0 |
| ZoDiac | $p^* = 0.95$ | 0.032 | 0.994 | 0.994 | 0.994 | 0.996 | 0.998 | 0.998 | 0.998 | 0.998 | 0.998 |
| ZoDiac | $p^* = 0.90$ | 0.062 | 0.996 | 0.996 | 0.998 | 0.998 | 0.998 | 0.998 | 0.998 | 0.998 | 0.998 |

Table 9: Results on different attack strengths of the **Contrast** attack .

| | Detection Threshold | FPR | WDR under Contrast adjustment factor of | | | | | | | | |
|---|---|---|---|---|---|---|---|---|---|---|---|
| | | | 0.2 | 0.3 | 0.4 | 0.5 | 0.6 | 0.7 | 0.8 | 0.9 | 1.0 |
| PSNR | - | - | 14.54 | 17.50 | 23.03 | 28.62 | 30.56 | 33.05 | 36.57 | 38.57 | 100.0 |
| StegaStamp | 61/96 | 0.056 | 0.730 | 0.918 | 0.98 | 0.998 | 1.0 | 1.0 | 1.0 | 1.0 | 1.0 |
| StegaStamp | 60/96 | 0.094 | 0.768 | 0.956 | 0.998 | 1.0 | 1.0 | 1.0 | 1.0 | 1.0 | 1.0 |
| ZoDiac | $p^* = 0.95$ | 0.032 | 0.998 | 0.996 | 0.996 | 0.998 | 0.998 | 0.998 | 0.998 | 0.998 | 0.998 |
| ZoDiac | $p^* = 0.90$ | 0.062 | 0.994 | 0.996 | 0.998 | 0.998 | 0.998 | 0.998 | 0.998 | 0.998 | 0.998 |

Table 10: Results on different attack strengths of the **JPEG** attack.

| | Detection Threshold | FPR | WDR under JPEG quality setting of | | | | | | | | |
|---|---|---|---|---|---|---|---|---|---|---|---|
| | | | 10 | 20 | 30 | 40 | 50 | 60 | 70 | 80 | 90 |
| PSNR | - | - | 28.05 | 30.82 | 32.28 | 33.23 | 34.02 | 34.68 | 35.62 | 36.88 | 39.09 |
| StegaStamp | 61/96 | 0.056 | 0.758 | 0.996 | 1.0 | 1.0 | 1.0 | 1.0 | 1.0 | 1.0 | 1.0 |
| StegaStamp | 60/96 | 0.094 | 0.806 | 1.0 | 1.0 | 1.0 | 1.0 | 1.0 | 1.0 | 1.0 | 1.0 |
| ZoDiac | $p^* = 0.95$ | 0.032 | 0.746 | 0.956 | 0.982 | 0.986 | 0.992 | 0.994 | 0.994 | 0.994 | 0.996 |
| ZoDiac | $p^* = 0.90$ | 0.062 | 0.834 | 0.986 | 0.99 | 0.99 | 0.992 | 0.996 | 0.996 | 0.996 | 0.998 |

Table 11: Results on different attack strengths of the **G-Noise** attack.

| | Detection Threshold | FPR | WDR under G-Noise standard deviation of | | | | | | | | |
|---|---|---|---|---|---|---|---|---|---|---|---|
| | | | 0.02 | 0.03 | 0.04 | 0.05 | 0.06 | 0.07 | 0.08 | 0.09 | 0.1 |
| PSNR | - | - | 34.04 | 30.59 | 28.14 | 26.26 | 24.71 | 23.41 | 22.29 | 21.32 | 20.45 |
| StegaStamp | 61/96 | 0.056 | 1.0 | 1.0 | 1.0 | 0.998 | 0.998 | 0.998 | 0.998 | 0.998 | 0.996 |
| StegaStamp | 60/96 | 0.094 | 1.0 | 1.0 | 1.0 | 1.0 | 1.0 | 1.0 | 1.0 | 1.0 | 0.998 |
| ZoDiac | $p^* = 0.95$ | 0.032 | 0.998 | 0.998 | 0.998 | 0.996 | 0.996 | 0.996 | 0.996 | 0.996 | 0.994 |
| ZoDiac | $p^* = 0.90$ | 0.062 | 1.0 | 0.998 | 0.998 | 0.996 | 0.996 | 0.996 | 0.996 | 0.996 | 0.996 |

Table 12: Results on different attack strengths of the **G-Blur** attack.

| | Detection Threshold | FPR | WDR under G-Blur kernel size of | | | | | | | |
|---|---|---|---|---|---|---|---|---|---|---|
| | | | 5 | 7 | 9 | 11 | 13 | 15 | 17 | 19 |
| PSNR | - | - | 32.17 | 29.42 | 28.00 | 27.04 | 26.22 | 25.60 | 25.08 | 24.59 |
| StegaStamp | 61/96 | 0.056 | 1.0 | 1.0 | 0.998 | 0.998 | 0.998 | 0.998 | 0.996 | 0.99 |
| StegaStamp | 60/96 | 0.094 | 1.0 | 1.0 | 1.0 | 1.0 | 1.0 | 1.0 | 1.0 | 0.996 |
| ZoDiac | $p^* = 0.95$ | 0.032 | 0.996 | 0.994 | 0.994 | 0.994 | 0.994 | 0.994 | 0.994 | 0.992 |
| ZoDiac | $p^* = 0.90$ | 0.062 | 0.996 | 0.996 | 0.994 | 0.994 | 0.994 | 0.994 | 0.994 | 0.994 |

Table 13: Results on different attack strengths of the **Bmshij18** attack.

| | Detection Threshold | FPR | WDR under Bmshij18 quality setting of | | | | |
|---|---|---|---|---|---|---|---|
| | | | 2 | 3 | 4 | 5 | 6 |
| PSNR | - | - | 30.70 | 32.26 | 33.90 | 35.44 | 37.17 |
| StegaStamp | 61/96 | 0.056 | 0.996 | 0.998 | 0.998 | 1.0 | 1.0 |
| StegaStamp | 60/96 | 0.094 | 0.998 | 1.0 | 1.0 | 1.0 | 1.0 |
| ZoDiac | $p^* = 0.95$ | 0.032 | 0.986 | 0.986 | 0.99 | 0.99 | 0.992 |
| ZoDiac | $p^* = 0.90$ | 0.062 | 0.968 | 0.992 | 0.992 | 0.996 | 0.996 |

Table 14: Results on different attack strengths of the **Cheng20** attack.

| | Detection Threshold | FPR | WDR under Cheng20 quality setting of | | | | |
|---|---|---|---|---|---|---|---|
| | | | 2 | 3 | 4 | 5 | 6 |
| PSNR | - | - | 31.79 | 33.22 | 35.07 | 36.58 | 37.98 |
| StegaStamp | 61/96 | 0.056 | 0.994 | 1.0 | 1.0 | 1.0 | 1.0 |
| StegaStamp | 60/96 | 0.094 | 0.996 | 1.0 | 1.0 | 1.0 | 1.0 |
| ZoDiac | $p^* = 0.95$ | 0.032 | 0.97 | 0.978 | 0.98 | 0.992 | 0.994 |
| ZoDiac | $p^* = 0.90$ | 0.062 | 0.984 | 0.986 | 0.99 | 0.99 | 0.994 |

Table 15: Results on different attack strengths of the **Zhao23** attack.

| | Detection Threshold | FPR | WDR under Zhao23 denoising steps of | | | | | | | | |
|---|---|---|---|---|---|---|---|---|---|---|---|
| | | | 40 | 60 | 80 | 100 | 120 | 140 | 160 | 180 | 200 |
| PSNR | - | - | 27.15 | 26.28 | 25.56 | 24.91 | 24.35 | 23.81 | 23.33 | 22.86 | 22.44 |
| StegaStamp | 61/96 | 0.056 | 0.588 | 0.286 | 0.098 | 0.032 | 0.01 | 0.002 | 0.002 | 0.0 | 0.0 |
| StegaStamp | 60/96 | 0.094 | 0.674 | 0.386 | 0.144 | 0.06 | 0.024 | 0.004 | 0.002 | 0.0 | 0.0 |
| ZoDiac | $p^* = 0.95$ | 0.032 | 0.98 | 0.974 | 0.95 | 0.94 | 0.924 | 0.912 | 0.894 | 0.866 | 0.818 |
| ZoDiac | $p^* = 0.90$ | 0.062 | 0.988 | 0.988 | 0.964 | 0.97 | 0.952 | 0.946 | 0.938 | 0.926 | 0.898 |

## B.7 Injecting Watermark into Frequency Space *VS* Spatial Space

We compare adding watermarks in the spatial domain of the noisy latent, denoted as ZoDiac w/o FT, where FT stands for Fourier Transformation, to our frequency-domain-based approach. As illustrated in Table 16, operating in the spatial domain results in a degraded PSNR, underscoring the importance of utilizing the frequency domain to maintain image quality. More significantly, *we observe distortions that are visible to the naked eye in the central region of the image*, as shown in Figure 10. This indicates that adding watermarks directly to the spatial domain of the noisy latent representation leads to conspicuous patterns in the generated watermarked images, thereby achieving robust watermarking at the expense of unacceptable visual artifacts.

Table 16: The watermarked image quality and watermark robustness in terms of watermark detection rate (WDR) before and after attack for ZoDiac and its variant ZoDiac w/o FT. The SSIM threshold is set to $s^* = 0.92$.

| Method | Image Quality | | | Watermark Detection Rate (WDR) before and after attack | | | | | | | | | | |
| | PSNR | SSIM | LPIPS | Pre-Attack | Post-Attack | | | | | | | | | |
| | | | | | Brightness | Contrast | JPEG | G-Noise | G-Blur | BM3D | Bmshj18 | Cheng20 | Zhao23 | Rotation |
| ZoDiac w/o FT | 26.47 | 0.92 | 0.09 | 1 | 1 | 1 | 1 | 1 | 1 | 1 | 1 | 1 | 1 | 0.998 |
| ZoDiac | 29.41 | 0.92 | 0.09 | 0.998 | 0.998 | 0.998 | 0.992 | 0.996 | 0.996 | 0.994 | 0.992 | 0.986 | 0.988 | 0.538 |

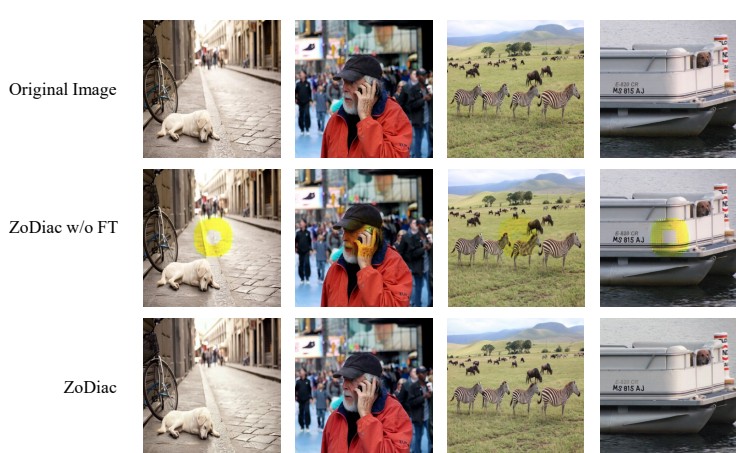

Figure 10: Visual comparisons between watermarked images generated by ZoDiac and its variant ZoDiac w/o FT.

## B.8 Discussion on Composite Attack

When performing composite attacks, the order of executing individual attacks may influence the watermarked detection rate. We evaluate with another composite attack "All w/o ↻ Rev.", which is the reversed order of "All w/o ↻ " in Table 17. The slightly decreasing WDR of *All w/o ↻ Rev.*, from $0.510 - 0.548$ to $0.456 - 0.490$, indicates that the order of the attack execution also plays a role in attacking watermarked images. It is worthwhile to deeply investigate composite attacks and develop image watermarking methods robust to various cases in the future.

Table 17: ZoDiac's watermark detection rate (WDR) two composite attacks, "All w/o ↻ " that combines all the individual attacks but excludes the rotation, and "All w/o ↻ Rev." that in the reversed order.

| Attack | ZoDiac Watermark Detection Rate (WDR)↑ | | |
| | MS-COCO | DiffusionDB | WikiArt |
| All w/o ↻ | 0.510 | 0.548 | 0.530 |
| All w/o ↻ Rev. | 0.490 | 0.508 | 0.456 |

## C More Attack Discussion

### C.1 About Rotation Attack

The rotation attack presents a significant challenge for most existing watermarking methods, including ZoDiac (see Table 1). There are two strategies to mitigate this issue. The first approach, exemplified by SSL, integrates rotation into the model's training phase as a type of data augmentation. However, this method is incompatible with ZoDiac's zero-shot nature. The second approach involves rectifying the image's orientation prior to watermark detection. This correction can be achieved manually or via an automated process by systematically rotating the image through a series of predetermined angles, spanning the full 360-degree spectrum and performing watermark detection at each increment. We call this the "rotation auto-correction" defense.

In this study, we test rotation auto-correction with ZoDiac. We conduct experiments using intervals of 5, 10, and 30 degrees and report the Watermark Detection Rate (WDR) for watermarked images subjected to the 90-degree rotation attack, and the False Positive Rates (FPR) for unmarked images. Table 18 reports the results on the MS-COCO dataset. As expected, applying rotation auto-correction enables ZoDiac to achieve a higher WDR. However, it also increases the risk of detecting a watermark in unmarked images, leading to a higher FPR. A higher detection threshold $p^* = 0.99$ can greatly suppress FPR with a slight decrease in WDR, decreasing from $0.244 - 0.676$ to $0.014 - 0.034$. Therefore in practice, to deploy the rotation auto-correction defense, the higher detection threshold is appropriate.

Table 18: ZoDiac's watermark detection rate (WDR) and false positive rate (FPR) under *Rotation* attack, with automatic correction of image orientation.

| MS-COCO, $s^* = 0.92$ | Detection Threshold $p^*$ | WDR | FPR |
|---|---|---|---|
| No Correction | 0.9 | 0.538 | 0.062 |
| | 0.99 | 0.376 | 0.004 |
| Correction Rotation Step Size — 5 | 0.9 | 0.998 | 0.676 |
| | 0.99 | 0.992 | 0.034 |
| Correction Rotation Step Size — 10 | 0.9 | 0.998 | 0.432 |
| | 0.99 | 0.992 | 0.016 |
| Correction Rotation Step Size — 30 | 0.9 | 0.998 | 0.244 |
| | 0.99 | 0.992 | 0.014 |

### C.2 About Pipeline-Aware Attack

Conducting a pipeline-aware attack involves simulating what a knowledgeable attacker would do if they were aware of how our watermarking works. The goal is to evaluate the robustness of ZoDiac against real-world threats where attackers might have insights into the defense strategies used. Specifically, we assume the attacker can inject another watermark to potentially disrupt or overwrite the original one by using ZoDiac, while testing whether the original watermark can still be preserved.

Table 19 reports the Watermark Detection Rate (WDR) for both the original and newly injected watermarks under four scenarios according to the extent of information available to the attacker. When the attacker has only partial knowledge of our watermarking method, such as lacking the model weights (i.e., the second row) or being unaware of the watermark injection configurations, including the injection channel and watermark radius (i.e., the third row), or both (i.e., the first row), the original watermark remains undisturbed by the new one and it is successfully detected with a high WDR. However, when the attacker has complete knowledge of the watermarking pipeline used for injecting the original watermark (i.e., the fourth row), the original watermark is unfortunately overwritten by the new one. The results confirm the robustness of our proposed ZoDiac in the face of partial knowledge adaptive attacks and also underscore the critical importance of safeguarding the model weights and watermark configurations to ensure security.

Table 19: The Watermark Detection Rate (WDR) for both the original and newly injected watermarks under four scenarios according to whether the attacker knows the model weights and the injection configurations of the original watermark.

| Model Weights | Watermark Setting | Original WDR | New WDR |
|---|---|---|---|
| × | × | 0.998 | 0.952 |
| × | √ | 0.992 | 0.998 |
| √ | × | 0.998 | 0.98 |
| √ | √ | 0.106 | 0.998 |

## D GPU Usage and Time Cost

The GPU usage of ZoDiac is 15,552MiB when operating with 32-bit floating-point precision. This memory is primarily attributed to the activation maps and model parameters required for the diffusion model computations, where the model parameters take 5,308MiB memory. All the experiments are conducted with 1 NVIDIA RTX8000.

Since ZoDiac is a zero-shot watermarking method based on the well-trained diffusion model, the time cost is solely attributed to optimizing the noisy latent representation of the given image embedded with the watermark. Therefore the time cost is influenced by the selected denoising step during the reconstruction of the watermarked

image. The completion of reconstruction is determined when the SSIM loss falls below 0.2 (i.e., the SSIM metric for the watermarked image quality exceeds 0.8) or the maximum number of iterations (i.e., 100) is reached.

As shown in Table 20, the larger the denoising step, the longer each iteration will take, consequently increasing the overall time cost. Regarding average convergence iterations, as the denoising step decreases, ZoDiac requires slightly fewer iterations to achieve similar performance. This might be attributed to the shallower steps of gradient backpropagation, leading to faster convergence. Notice that when the denoising step is decreased to 1, it becomes harder to reconstruct a watermarked image with a quality above 0.8, making it more challenging to trigger the early exit condition based on the SSIM loss threshold. Consequently, it leads to longer convergence iterations in this case. Additionally, we have demonstrated that the denoising steps do not significantly impact the watermarking performance. The detailed results and analysis are provided in Appendix B.2. Therefore, we could always choose small denoising steps to effectively embed robust watermarks while maintaining a reasonable time cost.

Table 20: Average convergence iterations, time cost per iteration, and time cost per image when utilizing different denoising steps for watermarking one image.

| Denoising Steps | Avg. Iteration | Time Cost (s/iter) | Time Cost (s/Img) |
|---|---|---|---|
| 50 | 43.9 | 5.83 | 255.94 |
| 30 | 42.3 | 3.77 | 159.47 |
| 10 | 38.9 | 1.68 | 65.35 |
| 1 | 68.1 | 0.66 | 45.41 |

# E  Visual Examples

Figure 11 shows several image examples, taken from each of our three datasets. For each image, we show the original, watermarked versions with seven watermarking methods, including ZoDiac, and the residual images (i.e., the watermarked image minus the original image). We observe that ZoDiac does not introduce significant visible degradation. We can hardly observe image differences for the DwtDct, RivanGAN, and SSL methods, a small amount of differences for DwtDctSvd, CIN, and our proposed ZoDiac, and a significant amount for StegaStamp. Notably, DwtDct, RivanGAN, and SSL suffer from poor robustness against advanced attacks, while ZoDiac achieves a favorable trade-off between visual quality preservation and robust watermarking.

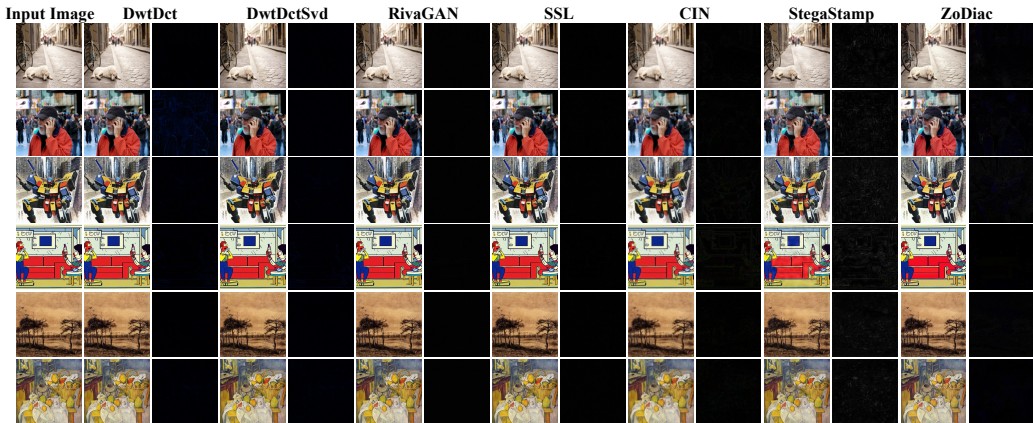

Figure 11: Visual examples of different invisible watermarking methods together with corresponding residual images (zoom in for details). The top two rows show real photos from the MS-COCO dataset, the next two rows show AI-generated content from DiffusionDB, and the last two rows are artworks from WikiArt. The SSIM threshold for ZoDiac is 0.92 for these examples.

# F  Impact Statement

Our work advances the robust image watermarking research field. In particular, we produce the first diffusion-model-based image watermarking method that is resilient to advanced watermark removal attacks, such as a diffusion-model-based attack and attacks composed of multiple, individual attack methods. Our method enables content protection and authenticity verification, positively impacting visual artists and other visual content creators. Our technique may use machine learning and is thus susceptible to some of the pitfalls of machine learning, such as bias, which may result in negative societal impact.

