# OpenReview forum: "Attack-Resilient Image Watermarking Using Stable Diffusion"
_NeurIPS.cc/2024/Conference — NeurIPS 2024 poster_

### Official Review · Reviewer_AMWU · 2024-06-25

**Soundness:** 3
**Presentation:** 3
**Contribution:** 3
**Rating:** 7
**Confidence:** 5

**Summary:**

This paper presents a new image watermarking method, ZoDiac, which leverages pretrained stable-diffusion model to generate watermark in the latent space (derived by using DDIM to process the original image). The author claims that ZoDiac achieves better robustness than the state-of-the-art (SOTA) watermarking methods.

**Strengths:**

The idea proposed in this work is original. The paper also introduces the proposed method clearly and with sufficient details.

**Weaknesses:**

Lack of sufficient details for the benchmark methods (both other watermark competitors and the attack methods), making it questionable if the comparison of robustness is fair and significant; the experiments and the logic conveyed in this paper is not convincing enough to support the claim that ZoDiac is more robust than the existing SOTA method, i.e.., StegaStamp. Please see “Questions” below for detailed comments.

**Questions:**

1. Are all watermarking methods compared fairly?

As stated in line 238 and 252, the watermark detection for the proposed ZoDiac relies on a threshold of (1-p). Analogously, for all other watermark competitors considered in this paper (DwtDctSvd, RivaGAN, SSL, CIN, StegaStamp), the watermark detection relies on thresholding the bitwise accuracy (or bit-error-rate in CIN) between the decoded bitstring and the ground truth. Altering these thresholds will lead to different trade-off points of True-Positive-Rate (TPR) and False-Positive-Rate (FPR) in the watermark detection performance.

The authors seem to be aware of this trade-off as in line 252 they report the detection threshold for ZoDiac; in line 11 they stress that ZoDiac has more than 98% TPR and less than 6.4% FPR; and in Table 3 they present an ablation study for ZoDiac. However, the detection threshold and FPR number for other watermarking methods are never mentioned in this paper (including Appendix). Imagine if another method (let’s call it B) achieves 97% TPR and 3% FPR, is ZoDiac or B more robust? As a result, I do not think the current evaluation is fair to compare the robustness of all watermarking methods.

Considering the fact that the detection threshold for each method may not be comparable, it would be reasonably fairer to make the following comparison to demonstrate robustness—let all methods achieve the same TPR then compare their FPR; or vice versa. In fact, the best demonstration here would be profiling the TPR-FPR curve for all watermarking methods to show the whole picture.

2. Has the definition of robustness clearly explained in this paper?

According to line 94-96, the robustness is defined as “watermark being detectable in attacked images” and attacked images should be “without significantly changing the image”. Have the authors defined and justified what is “significant”? Without properly addressing the question of “significance”, the robustness is essentially a trade-off between three quantities: (1) TPR, (2) FPR, and (3) attack image quality (inverse significance). Imagine that the attacked images are all with very severe additive gaussian noise such that no watermark can be correctly detected by any method considered, can we conclude that all watermarks are equally (non)robust? An example of justification can be found in [1], e.g., the contribution (1) and (2) mentioned. This may not be perfect but explains why [1] performs the robustness evaluation as Fig 3. I suggest the authors carefully consider this question, make the justification of their consideration and adjust their evaluation experiment accordingly, which may change/bring new insights to their robustness conclusions.

3.  Can you justify the selection of the attack methods used in testing the robustness? What is the criterion/consideration in selecting these attacks?

All the benchmark attacks considered in this paper (Brightness, Contrast, JPEG, Gaussian noise, Gaussian blur, BM3D, Bmshj18, Cheng20, Zhao23) have tunable parameters (e.g., corruption level, standard deviation, compression rate, kernel size, etc.) to adjust the attack strength. However, the author did not provide any information on what levels of these attacks are used in the experiment. Altering these parameters will inevitably affect the TPR-FPR-imageQuality trade-off. It is essential for the authors to provide details of the choice of these parameters and justify how it is related to the robustness considered in this paper.

[1] An, Bang, et al. "WAVES: Benchmarking the Robustness of Image Watermarks." Forty-first International Conference on Machine Learning.

**Limitations:**

N/A. There's no possible negative social impact that needs to discuss explicitly.

---

> ### Author Rebuttal · Authors · 2024-08-07
>
> Thank you for your constructive comments and critical feedback. Below we provide one-to-one responses and detailed discussions.
>
> **Q1: Are all watermarking methods compared fairly?**
>
> **A1:** Please refer to the global response to all reviewers. Thank you!
>
> **Q2: Has the definition of robustness clearly explained in this paper?**
> **Q3: Can you justify the selection of the attack methods used in testing the robustness?**
>
> **A2&3:** We address these two questions together as they are closely related.
>
> We define watermark robustness as **the ability to detect the encoded watermark in images after they have been subjected to malicious attacks**, as mentioned in lines 94-96. Consider a real scenario involving the use of image watermarking techniques. An image owner will watermark their work before publishing it online. Malicious actors may then attempt to remove the watermark and republish the image. **The goal of these attacks is to maintain the image quality within an acceptable range of the published watermarked images while removing the watermark.**
>
> Consequently, when evaluating the robustness of watermarking methods, **we selected attack methods that do not cause significant image changes, as measured by PSNR and SSIM metrics relative to the watermarked images before the attack.** Specifically, we adopted the attack methods used in [1] as mentioned in line 264. These attacks degrade the image slightly but still maintain acceptable PSNR (above 25dB) and SSIM (above 0.7) levels (refer to Table 1 in [1]).
>
> For your reference, we provide the detailed attack settings below and will include these details in the paper:
> - Adjustments in brightness or contrast with a factor of 0.5.
> - JPEG compression with a quality setting of 50.
> - Image rotation by 90 degrees.
> - Addition of Gaussian noise with a std of 0.05.
> - Gaussian blur with a kernel size of 5 and std of 1.
> - BM3D denoising algorithm with a std of 0.1.
> - Two VAE-based image compression models, Bmshj18 and Cheng20 with compression factors of 3.
> - A stable diffusion-based image regeneration model, Zhao23 with 60 denoising steps.
>
> In Table 1, we also present the image quality of the attacked watermarked images on the MS-COCO dataset in our experiments. Our results indicate that the image quality of the attacked watermarked images is not significantly degraded compared to the watermarked versions. An exception is the image rotation by 90 degrees, which results in lower PSNR and SSIM due to the rotated content; however, this case is still worth discussing as it presents unique challenges, as detailed in line 300-305 and Appendix Section C.1.  In short, **we expect a robust watermarking method to detect the watermark from these slightly degraded images.**
>
> **Table 1. The image quality of the attacked watermarked images compared to those without being attacked.**
> | Attack Methods | PSNR (&uarr;) | SSIM (&uarr;) |
> |:--------------:|:-------------:|:-------------:|
> |   Brightness   |     21.89     |      0.73     |
> |    Contrast    |     28.62     |      0.83     |
> |      JPEG      |     34.02     |      0.92     |
> |     G-Noise    |     26.26     |      0.75     |
> |     G-Blur     |     32.17     |      0.92     |
> |      BM3D      |     35.72     |      0.92     |
> |     Bmshj18    |     32.26     |      0.89     |
> |     Cheng20    |     33.22     |      0.90     |
> |     Zhao23     |     26.28     |      0.91     |
> |    Rotation    |      9.90     |      0.28     |
>
> [1] Zhao, Xuandong, et al. "Invisible image watermarks are provably removable using generative ai." arXiv 2023.

---

> ### Comment · Reviewer_AMWU · 2024-08-07
> **Reply to the authors**
>
> I appreciate the authors' effort in the rebuttal. However, the response does not sufficiently address my concerns. Here are the reasons:
>
> >**1. How is table 1 (image quality of the attacked watermarked image) related to your claim "image quality within an acceptable range"? Or, what is an acceptable range?**
>
> If I take it as "introducing changes to the images that are undetectable by human eyes", I am pretty sure a number of perturbations tested (e.g., brightness, G-Noise, maybe more) can be visibly detected. If I take it as PSNR $\ge 20$, then the question is why PSNR $= 20$ is a reasonable cutoff. Can you further clarify your argument and make it rigorous?
>
> >**2. I do not see how the provided response support a "fair comparison".**
>
> **First**, comparison at the same watermark detection threshold is hardly ever a fair strategy. Let me give you an example with two watermarking systems, where both rely on thresholding the bitwise accuracy (with some abuse of notation, let's use $I$ as un-watermarked image, $I_w$ watermarked images and $D_i$ as the decoder):
>
> 1. System (1):   $D_1(I) = 0, ~ \forall I$ and $D_1(I_w) = 1, ~ \forall I_w$
> 2. System (2):   $D_2(I) = 0.4, ~ \forall I$ and $D_2(I_w) = 0.6, ~ \forall I_w$
>
> System (1) and (2) essentially provide the same performance (if measured by TPR-FPR curve for example). Will setting a threshold $p=0.01$ and compare reach the same conclusion?
>
> The above example only uses bitwise acc., not to mention that the proposed method "ZoDiac" is thresholding on a template matching distance.
>
> **Second**, comparison at equivalent FPR levels is not correct. The only rigorous way in this direction is to tune the threshold $p$ of each method and achieve the exact **same FPR** level (control of variables), then TPR (not only for $I$, but also $I_w$ under different attacks) can be meaningfully compared. If we want to cross-compare the performance among different attacks, the **level of attack** (e.g., image quality loss) **should also be explicitly controlled to the same level**. Otherwise, they are not cross-comparable as my comment in the previous iteration.
>
> **Third**, why the numbers in Table: **False Positive Rate on original images and Watermark Detection Rate (WDR)** can be meaningfully sum and average to come up with metric Avg. WDR?
>
> For example, I can make a counter-argument that (1) rotation, contrast, G-noise and Zhao23 introduce visible corruptions by human eyes and should not be considered; (2) Bmshj18 and Cheng20 are essentially similar methods and should be combined as one; etc. Of course, my argument above is debatable and not convincing, but so as yours at current level. Can you provide further logical justification why this is a valid metric? Otherwise, it falls into my comment above (in "Second").
>
> **Fourth**, if we take one step back and acknowledge that no single watermark currently outperforms the others under every attack, would you agree that showing and discussing the difference in behavior (e.g., XX is better under G-Noise but worse under Cheng20) is more sensible than trying to get a score and claim the overall superiority?
>
> **Lastly**, I acknowledge the paper [1] the authors pointed to as their major support of "fair" comparison setups. Unfortunately, it is not a published work in any peer-reviewed conferences yet and it also suffers the non-rigorous evaluation setups as well. It does not sufficiently justify the evaluation approach you have taken in this paper. I highly recommend the authors to [2] and consider how they evaluated the watermarking systems and why.
>
> [1] Zhao, Xuandong, et al. "Invisible image watermarks are provably removable using generative AI." arXiv 2023.
>
> [2] An, Bang, et al. "WAVES: Benchmarking the Robustness of Image Watermarks." Forty-first International Conference on Machine Learning.

---

> > ### Author Response · Authors · 2024-08-08
> > **Response to the reviewer (1/2)**
> >
> > Thank you for the prompt review and feedback.
> >
> > **For Q1**, we acknowledge that we followed the attack settings in [1], which was published in the ICML23 Workshop, instead of a peer-reviewed conference. To better understand "image quality within an acceptable range," we refer to the PSNR guidelines in [2], where PSNR < 20 is deemed unacceptable. Our experiment so far ensures the PSNR of the images after the attack to be larger than 20dB. We will add experiments that vary the parameters of the watermark attack methods to provide a more comprehensive analysis on  attacked image with different PSNR quality.
> >
> > [1] Zhao, Xuandong, et al. "Invisible image watermarks are provably removable using generative ai." arXiv 2023. OR Zhao X, Zhang K, Wang Y X, et al. Generative autoencoders as watermark attackers: Analyses of vulnerabilities and threats. 2023.
> > [2] Sara U, Akter M, Uddin M S. Image quality assessment through FSIM, SSIM, MSE and PSNR—a comparative study. Journal of Computer and Communications, 2019, 7(3): 8-18.

---

> > > ### Author Response · Authors · 2024-08-08
> > > **Response to the reviewer (2/2)**
> > >
> > > **For Q2**, we apologize for not including the TPR-FPR curve in the rebuttal due to limited time. However, **we believe we have formed a fair comparison, and the results show that ZoDiac demonstrates better robustness compared to the baselines, especially against advanced watermark attack methods.**
> > >
> > > To ensure clarity, let's first define the terms TPR, FPR and the Avg. WDR for consistent understanding:
> > > 1. **TPR/WDR**: The watermark detection rate on watermarked images. Higher values are better. When the watermark detection threshold increases (i.e., is more tight), TPR decreases.
> > > 2. **FPR**: The watermark detection rate on clean images. Lower values are better. When the watermark detection threshold increases, FPR decreases.
> > > 3. **Avg. WDR**: The averaged watermark detection rate aross all the attack methods, considering the evaluation importance of the different attacks are the same. Notice that we didn’t sum with FPR on the original images.
> > >
> > > Table 1 has shown, ZoDiac achieves lower FPR with a higher TPR than alternative approaches on many attack methods, implying that ZoDiac can achieve even higher TPR under the same FPR level. We will use the following example to elaborate on the statement. .
> > > -  **DwtDct vs ZoDiac**:
> > >    - When DwtDct’s FPR is 0.052, its WDR under Zhao23 attack is 0 and its highest WDR is achieved under G_noise attack which is 0.687.
> > >    - When ZoDiac’s FPR is 0.03, its WDR under Zhao23 attack is 0.974 and its WDR under G_noise attack is 0.996.
> > >    - Implication: Decreasing the detection threshold for ZoDiac will increasing both FPR and TPR. The WDR of ZoDiac will be even higher than that of DwtDct under the same FPR level against all attack methods.
> > > - **DwtDctSvd vs ZoDiac**:
> > >    - When comparing ZoDiac (with a FPR of 0.03) with DwtDctSvd (with a FPR of 0.018), ZoDiac significantly outperforms DwtDctSvd in WDR on 8 out of 10 attack methods with a 0.246-0.97 higher WDR.
> > >    - For the rest two attack methods (G-noise and G-Blur), ZoDiac achieves a slightly lower WDR with a maximum 0.004 difference. Increasing ZoDiac’s FPR to be of the same level as DwtDctSvd will further reduce the gap.
> > >
> > > We can get the conclusion for the rest of the methods in a similar way:.
> > > - **RivaGAN vs ZoDiac**: ZoDiac achieves 0.376-0.976 higher TPR against advanced methods (Bmshj18, Cheng20, Zhao23, Rotation) and competitive TPR against others.
> > > - **SSL vs ZoDiac**: ZoDiac shows 0.932-0.988 better TPR against JPEG, G-Noise, Bmshj18, Cheng20, and Zhao23 where SSL directly fails, and competitive results against Brightness, Contrast, and G-Blur. SSL outperforms ZoDiac in Rotation, which we analyze in lines 300-305 and Appendix C.1.
> > > - **CIN vs ZoDiac**: ZoDiac (with a FPR of 0.004) achieves 0.288-0.46 higher TPR than CIN (with a FPR of 0.026) against attack methods, JPEG, BM3D, Bmshj18, Cheng20, and Zhao23, and 0.01-0.11 lower TPR against other methods.
> > > - **StegaStamp vs ZoDiac**: ZoDiac (with a FPR of 0.03) demonstrates 0.688 higher TPR than StegaStamp (with a FPR of 0.056) under Zhao23 and 0.376 higher TPR under Rotation, with 0.004-0.012 lower TPR in other cases.
> > >
> > > In summary, the results in Table 1 provide sufficient evidence that ZoDiac demonstrates better robustness when facing different types of watermark removal attacks compared with the baselines.
> > > With that said, we believe that the paper could be further strengthened with more comparisons between ZoDiac and alternative approaches under (1) the exactly same FPR level by tuning detection threshold, and (2) different image quality degradations from attack methods by tuning attack strengths, as suggested by the reviewer. We will add these experiments in the revision.
> > >
> > > Finally, we agree with you that [3] is a very solid work in evaluating image watermarking approaches. We will refer to [3] when adding the above mentioned experiments. Fortunately, we reach to similar insights:  (1) there is always a tradeoff between the image quality of watermarked images and the watermark robustness and (2) StegaStamp achieves the strongest watermark robustness while introducing artifacts into images.
> > > Compared to StegaStamp, our proposed ZoDiac is more robust under the advanced watermark removal attack Zhao23 and achieves competitive WDR under other attacks as analyzed above. In terms of the image quality, ZoDiac has similar performance as StegaStamp as shown in Table 1 in the paper. From the visual comparison in Figure 11, we can observe a small amount of differences in the residual images for our proposed ZoDiac, and a significant amount for StegaStamp.
> > >
> > > [3] An, Bang, et al. "WAVES: Benchmarking the Robustness of Image Watermarks." Forty-first International Conference on Machine Learning.

---

> ### Comment · Reviewer_AMWU · 2024-08-08
> **Reply to the authors**
>
> Thank you for the prompt response. However, it seems to me that most of my concerns still remain. Please allow me to be brief:
>
> >1. About acceptable attack quality of image.
>
> The authors have pointed to reference [1]. However, I do not see how [1] can support the authors claim that "PSNR < 20 is deemed unacceptable" in the watermark application. Can the authors give any further logical argument how they arrive from [1] to this claim, for example, in terms of content protection in watermarking problems which has been mentioned as the motivation of this work in the introduction?
>
> >2. About **WDR**
>
> I do not have question how WDR is calculated. My concern is **why WDR is a valid** measure logically. That is to say, what is the logical argument that the watermark detection results from different attacks can be cross-comparable and averaged?
>
> For example, if "PSNR < 20 is deemed unacceptable" is the standard, why don't we tune the hyper-parameters of all attack methods to achieve the attack images with PSNR=20 first, then test the watermark detection performance using these attacked images and calculate the WDR?  To clarify, I do not mean that WDR is a sensible measure under this condition, but it seems to me a fairer setup than the authors have practiced.
>
> >3. About the detailed discussion/comparison of performance between ZoDiac and other existing methods provided by the authors in the reply (excluding content related to WDR).
>
> This detailed discussion is indeed sensible, e.g., identifying ZoDiac performs better under Zhao23 but worse on other more traditional image editing (such as jpeg, G-Noise), and is in fact more constructive and informative than trying to use WDR as a unified score and claim "the best". I would suggest the authors consider using these discussions in their main paper.
>
> >4. About suggesting following the evaluation proposed by [2]
>
> This suggestion is largely due to the unsettled logical augment of 1. and 2. Under this condition, faithfully showing the influence of robustness caused by image quality, FPR and TPR is necessary.
>
> Again, using the table at the top of the rebuttal (**False Positive Rate on original images and Watermark Detection Rate**) as an example.
>
> **(1)** Problems related to TPR/FPR point: without reporting different FPR point for other methods (e.g., StegaStamp of CIN), it is unclear if when FPR=0.062, these methods will have much more robust performance under attack.
>
> **(2)** Problems related to **image quality (generated by attack methods)**: if G-Noise attack level is doubled, it is unclear if ZoDiac (p=0.9) can maintain 0.996 detection rate. It would be unclear if the performance of ZoDiac will reduce to 0.5; what if StegaStamp can still maintain 1 at the same time? Similar concerns can be raised for all other attacks.
>
> Therefore, extra experiments are necessary to confirm the above cases are not happening and verify the robustness of the proposed method.
>
> I appreciate the authors acknowledgment and promises of adding the experiments that I suggested. Unfortunately, the rating can be only given to what is available in the submitted paper and rebuttal. The change of rating should be subject to timely addressing these concerns.
>
>
> [1] Sara U, Akter M, Uddin M S. Image quality assessment through FSIM, SSIM, MSE and PSNR—a comparative study. Journal of Computer and Communications, 2019, 7(3): 8-18.
>
> [2] An, Bang, et al. "WAVES: Benchmarking the Robustness of Image Watermarks." Forty-first International Conference on Machine Learning.

---

> > ### Author Response · Authors · 2024-08-09
> > **Response to the reviewer (1/3)**
> >
> > Thank you for your valuable suggestions. Below we provide our responses and additional experimental results to the key questions.
> >
> > **Q1:** How [1] can support the authors' claim that "PSNR < 20 is deemed unacceptable"
> >
> > **A1:** [1] mentioned that “In image and video compression quality degradation, the PSNR value varies from 30 to 50 dB for 8-bit data representation and from 60 to 80 dB for 16-bit data. In wireless transmission, the accepted range of quality loss is approximately 20 - 25 dB”.
> >
> > In wireless transmission, maintaining signal quality while allowing for some degradation is crucial for ensuring that transmitted data remains intelligible and useful. It is similar to the watermark attack methods whose goal is to remove the watermark while preserving the usability of content despite some level of degradation. Therefore, we believe that 20dB could be a reasonable PSNR threshold in our case.
> >
> > [1] Sara U, Akter M, Uddin M S. Image quality assessment through FSIM, SSIM, MSE and PSNR—a comparative study\[J]. Journal of Computer and Communications, 2019, 7(3): 8-18.
> >
> >
> > **Q2:** Why WDR from different attacks can be cross-comparable and why WDR is a valid measure? Why try to use Avg. WDR as a unified score and claim the best?
> >
> > **A2:** WDR is not used to compare across different attacks but to compare the robustness of different watermarking methods under the SAME attack. It is identical to TPR and we use WDR and TPR interchangeably in the rebuttal.  WDR/TPR is a commonly-used and valid metric.
> > The Avg. WDR provided in the global response is for reading the comparison more easily. We agree with you that it may not be a good choice to do averaging since the different attacks results in images with different image quality. **In fact, we didn’t use Avg. WDR in the result analysis in our paper and we clearly compare  ZoDiac with alternative watermarking approaches under each attack method in lines 293-299 in the paper.**

---

> > > ### Author Response · Authors · 2024-08-09
> > > **Response to the reviewer (2/3)**
> > >
> > > **Q3:** Without reporting different FPR points for other methods (e.g., StegaStamp of CIN), it is unclear if when FPR=0.062, these methods will have much more robust performance under attack.
> > >
> > > **A3:** We agree that comparisons under more FPR points would give a more comprehensive picture of how ZoDiac and alternative approaches perform.
> > >
> > > **First of all, we’d like to clarify that existing results already demonstrate the comparable robustness in terms of WDR, if not better, of ZoDiac compared to the alternative approaches under the considered FPR levels in Table 1 of the global response.** Note that the considered FPR levels for baselines are already higher than the FPR of 0.001 considered in WAVE [2] and the FPR of 0.01 in prior work [3]. This means that, if we decrease the FPR level of the baselines to be at the same level as ZoDiac (e.g., FPR=0.004), the WDR/TPR of the baselines would be even worse compared to ZoDiac. Even if we assume that their WDR/TPR will remain the same as we see in Table 1, we can still conclude that ZoDiac achieves 0.288-0.46 higher WDR/TPR than CIN against attack methods, JPEG, BM3D, Bmshj18, Cheng20, and Zhao23, and only 0.01-0.11 lower WDR/TPR against other methods. And ZoDiac demonstrates 0.688 higher WDR/TPR than StegaStamp under Zhao23 and 0.376 higher WDR/TPR under Rotation, with slightly (i.e., 0.004-0.012) lower WDR/TPR in other cases.
> > >
> > > Second, we acknowledge that Table 1 in the global response does not provide evidence for the advantage of ZoDiac over baselines on much higher FPR levels (e.g., FPR=0.062). **To help clarify the reviewer’s doubt, we conducted more experiments comparing ZoDiac with StegaStamp and CIN under a similar FPR of 0.062 as shown in Table 3 below**. Here we cannot get the exact same FPR of 0.062 since the threshold for CIN and StegaStamp are controlled by the number of correct bits, resulting in more coarse-grained FPR levels. Instead, we report the performance of CIN at FPR levels between 0.026-0.102 and StegaStamp between 0.056-0.094. We used the advanced attack approach Bmshj18, Cheng20, and Zhao23.
> > >
> > > Overall, ZoDiac still outperforms StegaStamp at FPR=0.094 against Zhao23 by 0.602 higher WDR, and outperforms CIN at FPR=0.102 against the three advanced attacks by 0.108-0.308 higher WDR. The result further confirms the advantages of ZoDiac over StegaStamp and CIN even at relatively high FPR levels.
> > >
> > > **Table 3. Comparison of WDR against the advanced attack methods under a similar high FPR of 0.062**
> > > |   Methods  | Detection Threshold / Accurate Bit Ratrio |  FPR  | Bmshj18 | Cheng20 | Zhao23 |
> > > |:----------:|:-----------------------------------------:|:-----:|:-------:|:-------:|:------:|
> > > |     CIN    |                   24/32                   | 0.026 |  0.662  |  0.666  |  0.478 |
> > > |     CIN    |                   23/32                   | 0.056 |  0.748  |  0.784  |  0.598 |
> > > |     CIN    |                   22/32                   | 0.102 |  0.848  |  0.878  |  0.680 |
> > > | StegaStamp |                   61/96                   | 0.056 |  0.998  |  1.000  |  0.286 |
> > > | StegaStamp |                   60/96                   | 0.094 |  1.000  |  1.000  |  0.386 |
> > > |   ZoDiac   |                 $p^*=0.90$                | 0.062 |  0.992  |  0.986  |  0.988 |
> > >
> > > [2] An, Bang, et al. "WAVES: Benchmarking the Robustness of Image Watermarks." Forty-first International Conference on Machine Learning.
> > > [3] Wen, Yuxin, et al. "Tree-ring watermarks: Fingerprints for diffusion images that are invisible and robust." NeurIPS 2023.

---

> > > > ### Author Response · Authors · 2024-08-09
> > > > **Response to the reviewer (3/3)**
> > > >
> > > > **Q4:** If "PSNR < 20 is deemed unacceptable" is the standard, why don't we tune the hyper-parameters of all attack methods to achieve the attack images with PSNR=20 first, then test the watermark detection performance using these attacked images and calculate the WDR?
> > > > **Q5:** Problems related to image quality (generated by attack methods): if G-Noise attack level is doubled, it is unclear if ZoDiac (p=0.9) can maintain a 0.996 detection rate. It would be unclear if the performance of ZoDiac will reduce to 0.5; what if StegaStamp can still maintain 1 at the same time? Similar concerns can be raised for all other attacks.
> > > >
> > > > **A4&5:** We agree that conducting additional experiments with more hyperparameter settings in attack methods could provide further empirical information on how ZoDiac and alternative approaches perform on attacked images with various image quality levels. Before we report some of these experiments in this direction to clarify the concerns, **we want to first emphasize that our current experiments already form a fair comparison and can faithfully reflect the robustness of ZoDiac.** Our experiment in the paper compares ZoDiac and alternative approaches against attacking methods (e.g., G-Noise) under the same hyper-parameter settings (e.g., Gaussian variance). After going through the SAME attack, our comparison assumes that the most robust watermarking method is the one that achieves the highest WDR/TPR while maintaining a low FPR, which is the same evaluation setting in many prior works including WAVES [2].
> > > >
> > > > To help clarify the reviewer’s concerns on how different image quality will affect the performance of ZoDiac and alternative approaches, we compare ZoDiac and the strongest baseline StegaStamp under different hyper-parameter settings in G-Noise. Specifically, **we add Gaussian random noise (i.e., the G-Noise attack) with a standard deviation varying from 0.02 to 0.1 following WAVE \[2] and compare the WDR of ZoDiac with StegaStamp**. Below we provide the image quality of the attacked images compared to the watermarked images before the attack in Table 4. Under the same attack hyperparameter setting, the image quality of the attacked images for ZoDiac and StegaStamp are basically the same. The FPR and WDR comparisons are in Table 5.
> > > >
> > > > **Table 4. The image quality of the attacked watermarked images compared to those without being attacked in terms of PSNR.**
> > > > | G-Noise std |  0.02 |  0.03 |  0.04 |  0.05 |  0.06 |  0.07 |  0.08 |  0.09 |  0.1  |
> > > > |:-----------:|:-----:|:-----:|:-----:|:-----:|:-----:|:-----:|:-----:|:-----:|:-----:|
> > > > |  StegaStamp | 34.00 | 30.55 | 28.10 | 26.21 | 24.67 | 23.38 | 22.26 | 21.29 | 20.42 |
> > > > |    ZoDiac   | 34.04 | 30.59 | 28.14 | 26.26 | 24.71 | 23.41 | 22.29 | 21.32 | 20.45 |
> > > >
> > > > **Table 5. The WDR of ZoDiac and StegaStamp under different strengths of the G-Noise attack.**
> > > > | G-Noise std | Detection Threshold / Accurate Bit Ratrio |  FPR  |  0.02 |  0.03 |  0.04 |  0.05 |  0.06 |  0.07 |  0.08 |  0.09 |  0.1  |
> > > > |:-----------:|:-----------------------------------------:|:-----:|:-----:|:-----:|:-----:|:-----:|:-----:|:-----:|:-----:|:-----:|:-----:|
> > > > |  StegaStamp |                   61/96                   | 0.056 |  1.0  |  1.0  |  1.0  | 0.998 | 0.998 | 0.998 | 0.998 | 0.998 | 0.996 |
> > > > |  StegaStamp |                   60/96                   | 0.094 |  1.0  |  1.0  |  1.0  |  1.0  |  1.0  |  1.0  |  1.0  |  1.0  | 0.998 |
> > > > |    ZoDiac   |                 $p^*=0.95$                | 0.032 | 0.998 | 0.998 | 0.998 | 0.996 | 0.996 | 0.996 | 0.996 | 0.996 | 0.994 |
> > > > |    ZoDiac   |                 $p^*=0.90$                | 0.062 |  1.0  | 0.998 | 0.998 | 0.996 | 0.996 | 0.996 | 0.996 | 0.996 | 0.996 |
> > > >
> > > > The results provide empirical evidence to answer Q4 and Q5 under the G\_noise attack:
> > > >
> > > > (1) Under the PSNR of 20 for attacked images, i.e., the G-Noise attack with std of 0.1, ZoDiac achieves competitive WDR with StegaStamp under similar FPR, which is consistent with the observations reported in the paper.
> > > >
> > > > (2) When G-Noise strength is doubled from std of 0.05 to 0.1, ZoDiac can still maintain its high WDR.

---

> > > > > ### Comment · Reviewer_AMWU · 2024-08-09
> > > > > **Reply to the authors**
> > > > >
> > > > > Thank you for the prompt response. After carefully reading your response and checked your original paper, I decide to temporarily raise the rating to **weak accept**. Below are the explanations:
> > > > >
> > > > > >About definition of robustness, acceptable attack level and authors' claim about the robustness of ZoDiac.
> > > > >
> > > > > It is still logically unclear how the PSNR threshold in multi-media compression/signal transmission can transfer to the application of watermarks. However, setting the acceptable level in practice (e.g., to what level of content preservation can be considered copyright protected) may not be a pure technical problem, I view this logical leap a secondary issue.
> > > > >
> > > > > In the original paper, the authors only claim that ZoDiac is a robust watermarking method, which I do agree. The avg. WDR in the rebuttal, however, did misled me with some biased impression of overclaiming. In this sense, the authors should get a fair credit based on the original paper content.
> > > > >
> > > > > >About all examples I provided above related to fair comparison about thresholding, FPR, image quality, attack type and level, etc.
> > > > >
> > > > > First of all, given all the experiment results so far (including ones in the paper and the additional ones in the rebuttal), I do not question the robustness of ZoDiac. The examples I gave above, however, is not for questioning the performance under a specific case. Instead, they are to explain why multiple detection thresholds (FPR/TPR cutoffs), attack levels are necessary to show more holistic pictures about robustness, which may in turn support the authors claim.
> > > > >
> > > > > >Summary of what affects the current rating.
> > > > >
> > > > > **Positives**: the proposed method is novel; claims are appropriate based on the current empirical evaluations.
> > > > >
> > > > > **Negatives**: more comprehensive evaluation is still possible and not covered by the authors, and it is possible that the authors' claims may be even strengthened after such evaluation.
> > > > >
> > > > > A further improvement on the rating is still possible if the holistic evaluation of robustness can be provided.

---

> ### Author Response · Authors · 2024-08-11
> **Response to the reviewer (1/3)**
>
> We really appreciate your decision and enjoy the discussion with you.
>
> In response to your suggestions, we conducted additional experiments under various attack levels, following the settings in WAVES, and will include these findings in the revised paper. The experiments span multiple attack scenarios:
> - **Tables 6 and 7**: Adjustments in brightness or contrast with a factor of 0.2 to 1.0 (original image).
> - **Table 8**: JPEG compression with a quality setting from 10 to 90
> - **Table 9**: Gaussian blur with a kernel size of 5 to 19.
> - **Tables 10 and 11**: Two VAE-based image compression models, Bmshj18 and Cheng20 with quality levels of 2 to 6.
> - **Table 12**: A stable diffusion-based image regeneration model, Zhao23 with 40 to 200 denoising steps.
>
> We compared ZoDiac and the strongest baseline, StegaStamp, in terms of WDR under different FPR levels, leading to several insights:
> - **Brightness and Contrast Attacks**: As the adjustment factors for brightness and contrast decrease from 0.9 to 0.2, the PSNR of the attacked watermarked images drops significantly—from 37.79 to 11.41 for brightness and from 38.57 to 14.54 for contrast. **Despite this degradation, ZoDiac consistently maintains a WDR above 0.99 across all scenarios.** Even at extreme settings, where PSNRs fall to 11.41 for brightness and 14.54 for contrast, ZoDiac significantly outperforms StegaStamp, achieving higher WDR and lower FPR. Specifically, ZoDiac records a WDR of 0.994 at an FPR of 0.032 compared to StegaStamp's 0.852 at an FPR of 0.056 for brightness, and a WDR of 0.998 at an FPR of 0.032 compared to StegaStamp's 0.730 at an FPR of 0.056 for contrast. **This clearly demonstrates ZoDiac’s robustness even under severe quality degradation.**
> - **JPEG, G-Blur, Bmshj18, and Cheng20 Attacks**: With varying attack parameters, including the quality settings for JPEG, Bmshj18, Cheng20, and the kernel size for G-Blur, the PSNR of the attacked watermarked images remains within the range of 25 to 40. **In these scenarios, both ZoDiac and StegaStamp consistently sustain a high WDR of around 0.98.** The only exception is JPEG with a quality setting of 10, where the PSNR drops to 28.05. Under this condition, the WDR of both ZoDiac and StegaStamp slightly decreases from their usual range of 0.956-0.986 to 0.746-0.834 for ZoDiac, and from 0.996-1.0 to 0.758-0.806 for StegaStamp.
> - **Zhao23**: As the number of image regeneration steps in Zhao23 increases, the PSNR of the attacked watermarked images decreases from 27.15 to 22.44. **Despite this degradation, ZoDiac remains highly resistant across most scenarios**, with its WDR slightly decreasing from 0.998 to 0.898. In contrast, StegaStamp significantly underperforms, completely failing when the regeneration steps exceed 120.
>
> In summary, the additional ablation study demonstrates that ZoDiac achieves either competitive or superior robustness performance compared with StegaStamp across a wide range of attack scenarios, highlighting its robustness and reliability even under severe conditions.

---

> ### Author Response · Authors · 2024-08-11
> **Response to the reviewer (2/3)**
>
> **Table 6.** The PSNR of attacked images and the WDR of ZoDiac and StegaStamp under different strengths of **the Brightness attack**.
> | Brightness factor | Detection Threshold |  FPR  |  0.2  |  0.3  |  0.4  |  0.5  |  0.6  |  0.7  |  0.8  |  0.9  |  1.0  |
> |:-----------------:|:-------------------:|:-----:|:-----:|:-----:|:-----:|:-----:|:-----:|:-----:|:-----:|:-----:|:-----:|
> |        PSNR       |          -          |   -   | 11.41 | 13.21 | 14.56 | 21.89 | 27.95 | 30.84 | 33.58 | 37.79 | 100.0 |
> |     StegaStamp    |        61/96        | 0.056 | 0.852 | 0.954 | 0.988 | 0.998 |  1.0  |  1.0  |  1.0  |  1.0  |  1.0  |
> |     StegaStamp    |        60/96        | 0.094 | 0.890 | 0.990 |  1.0  |  1.0  |  1.0  |  1.0  |  1.0  |  1.0  |  1.0  |
> |       ZoDiac      |      $p^*=0.95$     | 0.032 | 0.994 | 0.994 | 0.994 | 0.996 | 0.998 | 0.998 | 0.998 | 0.998 | 0.998 |
> |       ZoDiac      |      $p^*=0.90$     | 0.062 | 0.996 | 0.996 | 0.998 | 0.998 | 0.998 | 0.998 | 0.998 | 0.998 | 0.998 |
>
> **Table 7.** The PSNR of attacked images and the WDR of ZoDiac and StegaStamp under different strengths of **the Contrast attack**.
> | Contrast factor | Detection Threshold |  FPR  |  0.2  |  0.3  |  0.4  |  0.5  |  0.6  |  0.7  |  0.8  |  0.9  |  1.0  |
> |:---------------:|:-------------------:|:-----:|:-----:|:-----:|:-----:|:-----:|:-----:|:-----:|:-----:|:-----:|:-----:|
> |       PSNR      |          -          |   -   | 14.54 | 17.50 | 23.03 | 28.62 | 30.56 | 33.05 | 36.57 | 38.57 | 100.0 |
> |    StegaStamp   |        61/96        | 0.056 | 0.730 | 0.918 |  0.98 | 0.998 |  1.0  |  1.0  |  1.0  |  1.0  |  1.0  |
> |    StegaStamp   |        60/96        | 0.094 | 0.768 | 0.956 | 0.998 |  1.0  |  1.0  |  1.0  |  1.0  |  1.0  |  1.0  |
> |      ZoDiac     |      $p^*=0.95$     | 0.032 | 0.998 | 0.996 | 0.996 | 0.998 | 0.998 | 0.998 | 0.998 | 0.998 | 0.998 |
> |      ZoDiac     |      $p^*=0.90$     | 0.062 | 0.994 | 0.996 | 0.998 | 0.998 | 0.998 | 0.998 | 0.998 | 0.998 | 0.998 |
>
> **Table 8.** The PSNR of attacked images and the WDR of ZoDiac and StegaStamp under different strengths of **the JPEG attack**.
> | JPEG quality | Detection Threshold |  FPR  |   10  |   20  |   30  |   40  |   50  |   60  |   70  |   80  |   90  |
> |:------------:|:-------------------:|:-----:|:-----:|:-----:|:-----:|:-----:|:-----:|:-----:|:-----:|:-----:|:-----:|
> |     PSNR     |          -          |   -   | 28.05 | 30.82 | 32.28 | 33.23 | 34.02 | 34.68 | 35.62 | 36.88 | 39.09 |
> |  StegaStamp  |        61/96        | 0.056 | 0.758 | 0.996 |  1.0  |  1.0  |  1.0  |  1.0  |  1.0  |  1.0  |  1.0  |
> |  StegaStamp  |        60/96        | 0.094 | 0.806 |  1.0  |  1.0  |  1.0  |  1.0  |  1.0  |  1.0  |  1.0  |  1.0  |
> |    ZoDiac    |      $p^*=0.95$     | 0.032 | 0.746 | 0.956 | 0.982 | 0.986 | 0.992 | 0.994 | 0.994 | 0.994 | 0.996 |
> |    ZoDiac    |      $p^*=0.90$     | 0.062 | 0.834 | 0.986 |  0.99 |  0.99 | 0.992 | 0.996 | 0.996 | 0.996 | 0.998 |
>
> **Table 9.** The PSNR of attacked images and the WDR of ZoDiac and StegaStamp under different strengths of **the G-Blur attack**.
> | G-Blur kernel size | Detection Threshold |  FPR  |   5   |   7   |   9   |   11  |   13  |   15  |   17  |   19  |
> |:------------------:|:-------------------:|:-----:|:-----:|:-----:|:-----:|:-----:|:-----:|:-----:|:-----:|:-----:|
> |        PSNR        |          -          |   -   | 32.17 | 29.42 | 28.00 | 27.04 | 26.22 | 25.60 | 25.08 | 24.59 |
> |     StegaStamp     |        61/96        | 0.056 |  1.0  |  1.0  | 0.998 | 0.998 | 0.998 | 0.998 | 0.996 |  0.99 |
> |     StegaStamp     |        60/96        | 0.094 |  1.0  |  1.0  |  1.0  |  1.0  |  1.0  |  1.0  |  1.0  | 0.996 |
> |       ZoDiac       |      $p^*=0.95$     | 0.032 | 0.996 | 0.994 | 0.994 | 0.994 | 0.994 | 0.994 | 0.994 | 0.992 |
> |       ZoDiac       |      $p^*=0.90$     | 0.062 | 0.996 | 0.996 | 0.994 | 0.994 | 0.994 | 0.994 | 0.994 | 0.994 |

---

> ### Author Response · Authors · 2024-08-11
> **Response to the reviewer (3/3)**
>
> **Table 10.** The PSNR of attacked images and the WDR of ZoDiac and StegaStamp under different strengths of **the Bmshij18 attack**.
> | Bmshj18 quality level | Detection Threshold |  FPR  |   2   |   3   |   4   |   5   |   6   |
> |:---------------------:|:-------------------:|:-----:|:-----:|:-----:|:-----:|:-----:|:-----:|
> |          PSNR         |          -          |   -   | 30.70 | 32.26 | 33.90 | 35.44 | 37.17 |
> |       StegaStamp      |        61/96        | 0.056 | 0.996 | 0.998 | 0.998 |  1.0  |  1.0  |
> |       StegaStamp      |        60/96        | 0.094 | 0.998 |  1.0  |  1.0  |  1.0  |  1.0  |
> |         ZoDiac        |      $p^*=0.95$     | 0.032 | 0.986 | 0.986 |  0.99 |  0.99 | 0.992 |
> |         ZoDiac        |      $p^*=0.90$     | 0.062 | 0.968 | 0.992 | 0.992 | 0.996 | 0.996 |
>
> **Table 11**. The PSNR of attacked images and the WDR of ZoDiac and StegaStamp under different strengths of **the Cheng20 attack**.
> | Cheng20 quality level | Detection Threshold |  FPR  |   2   |   3   |   4   |   5   |   6   |
> |:---------------------:|:-------------------:|:-----:|:-----:|:-----:|:-----:|:-----:|:-----:|
> |          PSNR         |          -          |   -   | 31.79 | 33.22 | 35.07 | 36.58 | 37.98 |
> |       StegaStamp      |        61/96        | 0.056 | 0.994 |  1.0  |  1.0  |  1.0  |  1.0  |
> |       StegaStamp      |        60/96        | 0.094 | 0.996 |  1.0  |  1.0  |  1.0  |  1.0  |
> |         ZoDiac        |      $p^*=0.95$     | 0.032 |  0.97 | 0.978 |  0.98 | 0.992 | 0.994 |
> |         ZoDiac        |      $p^*=0.90$     | 0.062 | 0.984 | 0.986 |  0.99 |  0.99 | 0.994 |
>
> **Table 12.** The PSNR of attacked images and the WDR of ZoDiac and StegaStamp under different strengths of **the Zhao23 attack.**
> | Zhao23denoising step | Detection Threshold |  FPR  |   40  |   60  |   80  |  100  |  120  |  140  |  160  |  180  |  200  |
> |:--------------------:|:-------------------:|:-----:|:-----:|:-----:|:-----:|:-----:|:-----:|:-----:|:-----:|:-----:|:-----:|
> |         PSNR         |          -          |   -   | 27.15 | 26.28 | 25.56 | 24.91 | 24.35 | 23.81 | 23.33 | 22.86 | 22.44 |
> |      StegaStamp      |        61/96        | 0.056 | 0.588 | 0.286 | 0.098 | 0.032 |  0.01 | 0.002 | 0.002 |  0.0  |  0.0  |
> |      StegaStamp      |        60/96        | 0.094 | 0.674 | 0.386 | 0.144 |  0.06 | 0.024 | 0.004 | 0.002 |  0.0  |  0.0  |
> |        ZoDiac        |      $p^*=0.95$     | 0.032 |  0.98 | 0.974 |  0.95 |  0.94 | 0.924 | 0.912 | 0.894 | 0.866 | 0.818 |
> |        ZoDiac        |      $p^*=0.90$     | 0.062 | 0.988 | 0.988 | 0.964 |  0.97 | 0.952 | 0.946 | 0.938 | 0.926 | 0.898 |

---

> > ### Author Response · Authors · 2024-08-13
> > **Reponse to Reviewer AMWU**
> >
> > With the author-reviewer discussion period ending later today, we wanted to again thank you for a constructive discussion, and for helping us improve our manuscript.  We also would like to make a plea. We believe our paper accomplishes all important requirements to be accepted. Specifically, our paper (1) tackles an important problem, (2) makes considerable progress on that problem over the state of the art, (3) makes precise concrete claims that capture that progress, without overclaiming, and (4) supports those claims with empirical evidence.
> >
> > We believe from our discussion that you, too, agree that our paper accomplishes all four of those requirements and makes considerable progress on this important problem. **We would like to ask you to please consider supporting the acceptance of our paper more strongly**, as we believe its inclusion in the NeurIPS program will improve both the program and the state of science, and we hope you, too, agree with that statement.  Thank you!

---

> > > ### Comment · Reviewer_AMWU · 2024-08-14
> > > **Final rating updated**
> > >
> > > Dear authors,
> > >
> > > Your efforts are much appreciated. Most of my concerns have been addressed sufficiently. Therefore, my final recommendation for this paper has been updated as **accept**. If the paper is finally accepted, I hope the authors can find enough space to include what we all have agreed on.

---

> > > > ### Author Response · Authors · 2024-08-14
> > > > **Reponse to Reviewer AMWU**
> > > >
> > > > Thank you for your kind words and positive feedback. We are glad that most of your concerns have been addressed, and we greatly appreciate your final recommendation for acceptance. We will ensure that the agreed-upon results are included in the final version.

---

### Official Review · Reviewer_mbRv · 2024-07-07

**Soundness:** 3
**Presentation:** 3
**Contribution:** 3
**Rating:** 6
**Confidence:** 3

**Summary:**

The paper introduces ZoDiac, an image watermarking framework leveraging pre-trained stable diffusion models to embed concentric ring-like zero-bit watermark into existing images. ZoDiac operates by injecting the watermark into the trainable latent space. The method is extensively evaluated on three benchmarks MS-COCO, DiffusionDB, and WikiArt, demonstrating strong effectiveness and robustness, outperforming existing methods. ZoDiac is notable for its ability to watermark both AI-generated and real-world images without requiring extensive retraining, and it remains effective even against combined attack scenarios.

**Strengths:**

- The overall paper is easy to read and clear.
- Comprehensive experiments have been done to show effectiveness and robustness of ZoDiac against single or combinations of different attacks whereas several previous methods failed.
- ZoDiac does not need extensive training by leveraging pre-trained stable diffusion models, saving time and computational resources.

**Weaknesses:**

- The method seems inefficient as it needs 45.4 - 255.9s to watermark one image.
- While the paper demonstrates empirical robustness, it lacks theoretical support to explain why the approach is resilient to various attacks. The hypothesis about the reciprocating denoising process enhancing robustness of the watermark is not rigorously proven.
- The paper lacks experiments with watermark of different patterns apart from the concentric ring-like one.

**Questions:**

- Since previous works use different types of watermark, such as StegaStamp uses hyperlink bitstring, there is a concern regarding comparison fairness, hence the demonstrated results might be misleading. It would be better if the authors can provide clarification on this.
- It will be interesting to see if ZoDiac can work with watermark of different patterns.
- Other than UNet-based latent diffusion models like Stable Diffusion, it will be exciting to see if ZoDiac can work with Transformer-based ones like Pixart-alpha [1] or DiT [2].

[1]. Chen, J., Yu, J., Ge, C., Yao, L., Xie, E., Wu, Y., Wang, Z., Kwok, J., Luo, P., Lu, H., Li, Z.: PixArt-α: Fast training of diffusion transformer for photorealistic text-to-image synthesis. In: ICLR (2024)

[2]. William Peebles and Saining Xie. Scalable diffusion models with transformers. In ICCV, 2023.

**Limitations:**

The authors have adequately addressed the limitations and potential negative societal impact of their work.

---

> ### Author Rebuttal · Authors · 2024-08-07
>
> Thank you for your positive comments and rating! Below we provide one-to-one responses to the three questions.
>
> **Q1: Since previous works use different types of watermark, such as StegaStamp uses hyperlink bitstring, there is a concern regarding comparison fairness.**
>
> **A1:** Please refer to the global response to all reviewers. Thank you!
>
> **Q2: It will be interesting to see if ZoDiac can work with watermark of different patterns.**
>
> **A2:** The choice of a concentric ring-like pattern for our watermark is influenced by two key factors:
> (1) After frequency shifting, the center of the frequency map contains the low-frequency components, which can be slightly modified without significantly impacting the image content. This allows us to embed the watermark while maintaining the visual integrity of the image.
> (2) The frequency domain exhibits Hermitian symmetry, where the real part is even symmetric and the imaginary part is odd symmetric. This symmetry makes circularly symmetric watermarks the most straightforward option to align with this natural property.
> These considerations together led us to select the concentric ring-like watermark pattern. And we are excited about the potential to extend our approach to other patterns. Different patterns may open new possibilities for encoding longer and more meaningful information.
>
> **Q3: It will be exciting to see if ZoDiac can work with Transformer-based ones like Pixart-alpha or DiT.**
>
> **A3:** As requested, we have conducted additional experiments using the latest Transformer-based diffusion model, Huanyuan-DiT [1], and present the performance results in Tables 1 and 2 using MS-COCO dataset.
> Due to the time limit, the denoising step inside each optimization iteration for ZoDiac with DiT is set to 1 for the fastest watermarking on 50 images, and the SSIM threshold is set to 0.92 for fair comparison. Results from ZoDiac with other backbones are from Figure 9 in the paper and uses 50 iterations. **In summary, our method can seamlessly integrate with any stable diffusion backbone and achieve good performance.** The experimental results demonstrate the robustness of our approach, regardless of the specific diffusion model being employed.
>
> **Table 1. Watermarked image quality of ZoDiac with different backbones.**
> |       Backbone      | PSNR (&uarr;) | SSIM (&uarr;) | LPIPS (&darr;) |
> |:-------------------:|:-------------:|:-------------:|:--------------:|
> |      StegaStemp     |     28.64     |      0.91     |      0.13      |
> |  ZoDiac w/ 2.1base  |     29.41     |      0.92     |      0.09      |
> |    ZoDiac w/ 1.4    |     29.40     |      0.92     |      0.09      |
> | ZoDiac w/ XL1.0base |     29.41     |      0.92     |      0.09      |
> |    **ZoDiac w/ DiT**    |     28.55     |      0.92     |      0.13      |
>
> **Table 2. Watermark Detection Rate (WDR) (&uarr;) before and after attacking.**
> |       Backbone      | Pre-Attack | Brightness | Contrast |  JPEG | G-Noise | G-Blur |  BM3D | Bmshj18 | Cheng20 | Zhao23 | Rotation |
> |:-------------------:|:----------:|:----------:|:--------:|:-----:|:-------:|:------:|:-----:|:-------:|:-------:|:------:|:--------:|
> |      StegaStemp     |    1.000   |    0.998   |   0.998  | 1.000 |  0.998  |  1.000 | 0.998 |  0.998  |  1.000  |  0.286 |   0.000  |
> |  ZoDiac w/ 2.1base  |    0.998   |    0.998   |   0.998  | 0.992 |  0.996  |  0.996 | 0.994 |  0.992  |  0.986  |  0.988 |   0.538  |
> |    ZoDiac w/ 1.4    |    0.996   |    0.994   |   0.996  | 0.992 |  0.994  |  0.996 | 0.998 |  0.990  |  0.984  |  0.984 |   0.536  |
> | ZoDiac w/ XL1.0base |    0.998   |    0.998   |   0.998  | 0.992 |  0.994  |  0.998 | 0.996 |  0.992  |  0.986  |  0.986 |   0.536  |
> |  **ZoDiac w/ DiT**  |    1.000   |    0.998   |   0.998  | 0.996 |  0.996  |  0.996 | 0.996 |  0.992  |  0.990  |  0.988 |   0.538  |
>
> [1] Li, Zhimin, et al. "Hunyuan-DiT: A Powerful Multi-Resolution Diffusion Transformer with Fine-Grained Chinese Understanding." arXiv preprint arXiv:2405.08748 (2024).

---

> > ### Comment · Reviewer_mbRv · 2024-08-10
> >
> > I appreciate the authors' effective rebuttal. Most of my concerns have been addressed. However, my doubt regarding Q1 still remains. Since different methods use different types of message (32-bit messages for DwtDct, DwtDctSvd, RivaGAN, SSL, and CIN, and 96-bit messages for StegaStamp as stated), it is rather improper or misleading to compare them on the same scale as in Table 1 in the main paper without having any explanation or implication. I suggest to at least have some clarifications for them using different type of messages in the revision. Therefore, I keep my ratings unchanged.

---

> > > ### Author Response · Authors · 2024-08-11
> > > **Response to the reviewer**
> > >
> > > Thank you for your thoughtful feedback and for acknowledging our efforts in the rebuttal.
> > > We understand your concern about the comparison of different methods, particularly given the variations in message types. We will address this by adding explicit explanations in the revised paper, clarifying how the message types differ and discussing the implications of these differences in our comparisons.
> > >
> > > Thank you again for your constructive comments and for helping us improve our work.

---

> > > > ### Author Response · Authors · 2024-08-13
> > > > **Response to Reviewer mbRv**
> > > >
> > > > With the author-reviewer discussion period ending later today, we want to reach out to thank you again for your careful review and feedback. Regarding your concern about fair comparison with baselines encoding different types of messages, we would like to relay more justifications, which we will include in our manuscript.
> > > >
> > > > The fundamental problem we tackle is to **mark the images to claim ownership and effectively detect the encoded watermark in images even after being subjected to malicious attacks**. For this problem, solutions can use zero-bit or multi-bit watermarking methods.  Our evaluation compares robustness within the context of this problem in a fair way (as we explained in our global rebuttal).  Other problems may require multi-bit watermarks, but these are not the problems our paper tackles.  A single NeurIPS paper cannot do everything, and incorporating more complex watermarks while retaining ZoDiac's robustness to attacks is important future work. We are committed to making research progress in this direction.  Existing state-of-the-art evaluations, such as WAVE [1], also compare watermarking methods with different types of messages, including zero-bit method Tree-Rings and multi-bit method StegaStamp. We believe that we form a fair comparison following similar settings **for the specific problem our paper targets.**
> > > >
> > > > We believe our paper makes considerable progress on this important problem and hope you can support the acceptance of our paper. Publishing this significant advance in watermarking will enable future research into more complex watermarks.  Thank you again for recognizing our work's value.
> > > >
> > > > [1] An, Bang, et al. "WAVES: Benchmarking the Robustness of Image Watermarks." Forty-first International Conference on Machine Learning.

---

### Official Review · Reviewer_r9M9 · 2024-07-13

**Soundness:** 3
**Presentation:** 3
**Contribution:** 3
**Rating:** 5
**Confidence:** 3

**Summary:**

This paper presents ZoDiac, a novel image watermarking technique leveraging a pre-trained stable diffusion model to inject watermarks into a trainable latent space, enhancing watermark robustness against image attacks. Extensive experiments on three modern benchmarks demonstrate ZoDiac’s state-of-the-art watermark detection accuracy, outperforming existing methods in the face of traditional and generative model-based watermark removal attacks.

**Strengths:**

1. This paper is well-structured.
2. The authors claim that they have proposed for the first time an invisible watermark embedding framework that is robust to stable diffusion-based watermark removal attacks.
3. Extensive experiments have been conducted.

**Weaknesses:**

1. While ZoDiac demonstrates excellent robustness under various attacks, as shown in Table 1, it sacrifices image quality, achieving only slightly better visual quality than StegaStamp.

2. As the authors mentioned, the main weakness of the current method is that it is limited to zero-bit watermarking. How to increase the length of the embedded information is an issue that needs to be addressed.

**Questions:**

See the weaknesses.

**Limitations:**

N.A.

---

> ### Author Rebuttal · Authors · 2024-08-07
>
> Thank you for your positive comments and rating! Below we provide one-to-one responses to the two mentioned weaknesses.
>
> **W1: While ZoDiac demonstrates excellent robustness under various attacks, as shown in Table 1, it sacrifices image quality, achieving only slightly better visual quality than StegaStamp.**
>
> **A1:** We appreciate your observation regarding the trade-off between image quality and watermark robustness, which is a recognized challenge in watermarking techniques. Indeed, ZoDiac prioritizes watermark robustness, as shown in Table 1, which may result in a slight reduction in image quality. However, this reduction does not cause significant visible degradation, as evidenced by the corresponding watermarked images in Figure 11 in Appendix E. Furthermore, the trade-off curves in Figure 3 and Figures 6 to 8 in the Appendix also demonstrate that ZoDiac can even obtain comparable visual quality while achieving watermark robustness that is still on par with or surpasses existing state-of-the-art methods.
>
> **W2: As the authors mentioned, the main weakness of the current method is that it is limited to zero-bit watermarking. How to increase the length of the embedded information is an issue that needs to be addressed.**
>
> **A2:** We agree with you that extending ZoDiac to encode longer information is an exciting and valuable direction for future research. One potential approach is to develop an encoding-decoding mechanism that converts meaningful information into a Gaussian distribution, which can adhere to the distribution constraint in the latent vector of the diffusion model. We are committed to exploring this avenue and making progress in this area.

---

> > ### Author Response · Authors · 2024-08-13
> > **Reponse to Reviewer r9M9**
> >
> > With the author-reviewer discussion period ending later today, we wanted to reach out to thank you again for your careful review.
> >
> > ZoDiac uses zero-bit watermarking to solve important problems of tracking image provenance and proving ownership (as our introduction properly scopes).  There are other problems that require multi-bit watermarks that other papers have tackled, but a single NeurIPS paper cannot solve all problems related to watermarking. Incorporating more complex watermarks while retaining ZoDiac's robustness to attacks is important future work, but one paper cannot do everything. We conclusively demonstrate that ZoDiac significantly improves watermark robustness, while still achieving better image quality than the strongest baseline, StegaStamp. All watermarking methods sacrifice a little image quality -- that's the nature of the underlying problem.
> >
> > We are hopeful that you can support our paper’s acceptance. Publishing this significant advance in watermarking will enable future research into more complex watermarks.  Thank you for your help.

---

### Author Rebuttal · Authors · 2024-08-07

Thank you for your comments and suggestions. We first provide a general response to all reviewers with additional details.

**Q: Are all watermarking methods compared fairly? (Reviewer mbRv Q1, Reviewer AMWU Q1)**
**A:** Yes, our evaluation is designed to guarantee fair comparison with existing methods. We fully agree with the importance of performing fair comparisons and always keep this principle in mind when conducting experiments. To address your concerns comprehensively,  we'll elaborate on three key aspects.

**Q1.1: What are the threshold settings for the compared existing methods?**
**A1.1:** We adopted settings from a recent evaluation paper [1], using 32-bit messages for DwtDct, DwtDctSvd, RivaGAN, SSL, and CIN, and 96-bit messages for StegaStamp. Detection thresholds were set to reject the null hypothesis (i.e., $H_0$: the image has no watermark) with $p < 0.01$, requiring correct detection of 24/32 and 61/96 bits for the respective methods, as detailed in Section 2.3 in [1].

[1] Zhao, Xuandong, et al. "Invisible image watermarks are provably removable using generative AI." arXiv 2023.

**Q1.2: What is the False-Postive-Rate (FPR) of the compared existing methods?**
**A1.2:** Following your suggestion, we will extend Table 1 in the paper to include FPR for all the baselines and datasets. In summary, under the same FPR level, Zodiac always achieves a higher Watermark Detection Rate (WDR), demonstrating its superiority on watermark robustness. Please refer to the next question Q1.3 for detailed analysis.

Below we provide FPR (non-watermarked images detected as watermarked) and WDR (watermarked images detected as watermarked) before and after attacking for all the methods on the MS-COCO dataset. We also include our results with different watermark detection thresholds $p^*\in{0.90,0.95,0.99}$ in Table 5 from the Appendix to ease the comparison.

**Table 1. False Positive Rate (FPR, lower the better) on original images and Watermark Detection Rate (WDR, higher the better) on watermarked images before and after attacking. (Results on MS-COCO dataset)**
|   Watermarking Method   |  FPR  | Pre-Attack | Brightness | Contrast |  JPEG | G-Noise | G-Blur |  BM3D | Bmshj18 | Cheng20 | Zhao23 | Rotation | Avg. WDR |
|:-----------------------:|:-----:|:----------:|:----------:|:--------:|:-----:|:-------:|:------:|:-----:|:-------:|:-------:|:------:|:--------:|:--------:|
|          DwtDct         | 0.052 |    0.790   |    0.000   |   0.000  | 0.000 |  0.687  |  0.156 | 0.000 |  0.000  |  0.000  |  0.000 |   0.000  |   0.148  |
|        DwtDctSvd        | 0.018 |    1.000   |    0.098   |   0.100  | 0.746 |  0.998  |  1.000 | 0.452 |  0.016  |  0.032  |  0.124 |   0.000  |   0.415  |
|         RivaGAN         | 0.036 |    1.000   |    0.996   |   0.998  | 0.984 |  1.000  |  1.000 | 0.974 |  0.010  |  0.010  |  0.032 |   0.000  |   0.637  |
|           SSL           |   0   |    1.000   |    0.992   |   0.996  | 0.046 |  0.038  |  1.000 | 0.000 |  0.000  |  0.000  |  0.000 |   0.952  |   0.457  |
|           CIN           | 0.026 |      1     |      1     |     1    | 0.944 |    1    |    1   | 0.580 |  0.662  |  0.666  |  0.478 |   0.216  |   0.777  |
|        StegaStamp       | 0.056 |    1.000   |    0.998   |   0.998  | 1.000 |  0.998  |  1.000 | 0.998 |  0.998  |  1.000  |  0.286 |   0.000  |   0.843  |
| **ZoDiac -** $p^*=0.90$ | 0.062 |    0.998   |    0.998   |   0.998  | 0.992 |  0.996  |  0.996 | 0.994 |  0.992  |  0.986  |  0.988 |   0.538  |   **0.952**  |
| **ZoDiac -** $p^*=0.95$ | 0.030 |    0.998   |    0.996   |   0.998  | 0.992 |  0.996  |  0.996 | 0.994 |  0.986  |  0.978  |  0.974 |   0.376  |   **0.935**  |
| **ZoDiac -** $p^*=0.99$ | 0.004 |    0.992   |    0.99    |   0.99   | 0.978 |  0.984  |  0.988 | 0.988 |   0.96  |  0.954  |  0.938 |   0.106  |   **0.897**  |

**Q1.3: How do we form a fair comparison?**

**A1.3:** We ensure fair comparisons with all the existing watermarking methods by following two criteria:

**(1) Comparison at equivalent FPR levels:**
We compare WDR across different attack methods at the same FPR level. ZoDiac-$p^*=0.95$ outperforms DwtDct, RivaGAN, CIN, and StegaStamp with higher average WDR and lower FPR. When compared with DwtDctSvd and SSL, ZoDiac-$p^*=0.98$ consistently achieves higher WDR under competitive FPR, particularly against advanced watermark removal methods, Bmshj18, Cheng20, and Zhao23.

**(2) Comparison at the same watermark detection threshold:**
As described in A1.1, the decision threshold for existing methods is set to $p<0.01$ to reject the null hypothesis, which is equivalent to our $p^*=0.99$ threshold. Thus we compare ZoDiac-$p*=0.99$ with baselines. ZoDiac-$p^*=0.99$ maintains a higher average WDR while achieving an exceptionally low FPR of 0.004, highlighting the superiority of our method.

These comparisons demonstrate that ZoDiac offers robust performance across different attack methods and evaluation metrics, outperforming existing watermarking methods in challenging scenarios.

---

### Decision · Program_Chairs · 2024-09-25

**Decision:**

Accept (poster)

**Comment:**

This paper has received consistent feedback from all three reviewers. The reviewers engaged in thorough discussion and rebuttal, with most of their concerns being addressed, ultimately reaching a consensus. The paper proposes a framework for embedding invisible watermarks into existing images using a pretrained stable diffusion model. The experiments are comprehensive and effective. Therefore,  the AC  has decided to accept this paper.